# Understanding HTML with Large Language Models

## Abstract

Large language models (LLMs) have shown exceptional performance on a variety of natural language tasks. Yet, their capabilities for HTML understanding – i.e., parsing the raw HTML of a webpage, with applications to automation of web-based tasks, crawling, and browser-assisted retrieval – have not been fully explored. We contribute HTML understanding models (fine-tuned LLMs) and an in-depth analysis of their capabilities under three tasks: (i) *Semantic Classification* of HTML elements, (ii) *Description Generation* for HTML inputs, and (iii) *Autonomous Web Navigation* of HTML pages. While previous work has developed dedicated architectures and training procedures for HTML understanding, we show that LLMs pretrained on standard natural language corpora transfer remarkably well to HTML understanding tasks. For instance, fine-tuned LLMs are 12% more accurate at semantic classification compared to models trained exclusively on the task dataset. Moreover, when fine-tuned on data from the MiniWoB benchmark, LLMs successfully complete 50% more tasks using 192x less data compared to the previous best supervised model. Out of the LLMs we evaluate, we show evidence that T5-based models are ideal due to their bidirectional encoder-decoder architecture. To promote further research on LLMs for HTML understanding, we create and open-source a large-scale HTML dataset distilled and auto-labeled from CommonCrawl.[1]

## 1 Introduction

Web crawling (Olston et al., 2010), form-filling (Diaz et al., 2013; Gur et al., 2021), or information retrieving web agents (Nogueira & Cho, 2016) are important for both automating and assisting users in web-based tasks. These and similar applications rely on models that can search for specific content or controls on a web page as well as navigate a website autonomously. Since a web page in its raw form is represented as an HTML-based text sequence, the success of models for web-based tasks relies on their ability to understand HTML semantics, structure, and embedded interactions.

The predominant approach to web automation and HTML understanding is to train specialized models, i.e., gathering application-specific datasets and designing neural network (NN) architectures to leverage inductive biases of the HTML's structure; see, e.g., Liu et al. (2018); Toyama et al. (2021); Gur et al. (2021); Humphreys et al. (2022). However, both dataset collection and neural architecture design are expensive, time-consuming, and require highly-specialized, domain-specific knowledge.

Meanwhile, in the natural language processing (NLP) literature, large language models (LLMs) have emerged as a solution to the difficulties of dataset collection and specialized NN design (Kaplan et al., 2020; Bommasani et al., 2021). A popular paradigm in NLP is to take an off-the-shelf LLM – pretrained on a large text corpus via an unsupervised and task-agnostic learning objective – and either fine-tune or prompt the LLM on a small task-specific dataset. This paradigm has shown exceptional performance on a variety of NLP tasks (Xue et al., 2020; Brown et al., 2020; Austin et al., 2021). Whether LLMs can be applied to HTML understanding – especially given the much larger context and sequence lengths – remains an under-explored question.

In this paper, we investigate whether LLMs can be applied to HTML understanding to produce better-performing, more sample-efficient HTML understanding models and without the need for

---

[1]See visualizations of the results at `https://sites.google.com/view/llm4html/home`.

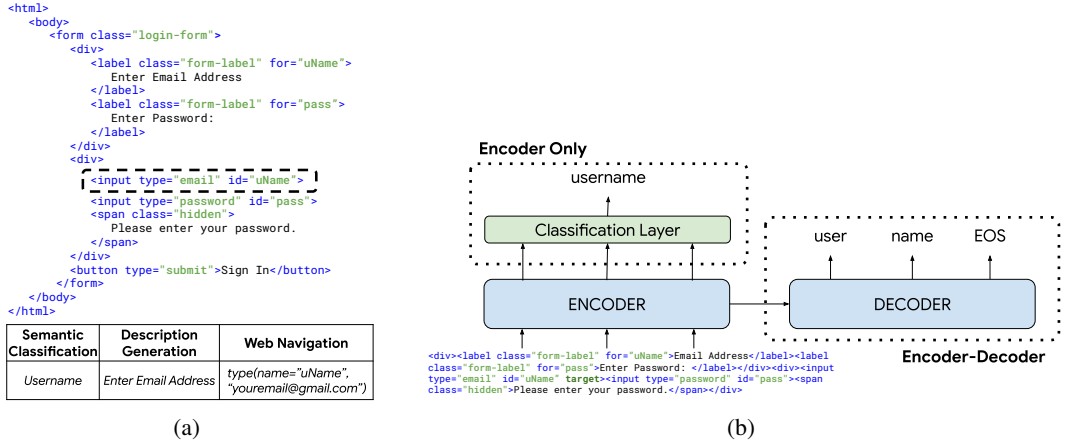

Figure 1: a) HTML example page with a highlighted salient element, an element of interest (dashed box). All canonical tasks evaluate a distinct interaction with this element, either by classifying it as one of a set of categories, generating a text description of its purpose, or applying an action as part of a sequential navigation of a multi-page website. b) LLM architectures overview. Dashed boxes denote sub-modules that are specific to either encoder-only or encoder-decoder models. For encoder-only models, we add an extra classification layer. Decoder-only models (not in the diagram) are similar to encoder-decoder models, the main difference is that the HTML snippet is fed to the decoder and processed from left-to-right.

custom NN architecture design. To that end, we present a suite of three benchmarking tasks for HTML understanding that capture the essence of these applications and require understanding both structure and content. First, we devise *Semantic Classification* as a task that requires a model to classify a given HTML element into one of a set of categories, such as address, email, password etc., with application to automated form-filling. Second, we present *Description Generation*, a label-extraction task where a model is given an HTML snippet and is asked to produce a natural language description. For instance for an email field, the description might be "Please enter your email address." Note that in the majority of web pages, this connection between input elements and description content is only implicit in the raw HTML code and inferring such links is a prerequisite for higher-level navigation objectives. The third task is *Autonomous Web Navigation* (Shi et al., 2017). A model is presented with an HTML page paired with a natural language command and must apply appropriate actions on a sequence of HTML pages to satisfy the command. See Figure 1a for a simplified example of these tasks.

With these benchmark tasks in hand, we evaluate the transfer capabilities of a variety of pretrained LLMs (Table 1), varying in architecture (encoder-only, encoder-decoder, or decoder-only), model size (from 24.6M to 62B parameters), and training data corpora (both including and excluding pre-training NLP and HTML corpus). While prior work universally pre-parses the HTML as input to the model (Gur et al., 2021; Liu et al., 2018; Nakano et al., 2021), ours – to the best of our knowledge – is the first work that uses raw, unprocessed HTML. Our results show that LLMs demonstrate a remarkable level of HTML understanding across all tasks, with up to $192\times$ more sample-efficiency than models trained from scratch, and achieving a new SoTA for supervised learning on the MiniWoB benchmark suite (Shi et al., 2017). The encoder-decoder architectures with bi-directional attention show the best performance across the board even when their pretraining does not include HTML. In addition, we show that the performance scales sub-linearly with the model size.

The broader objective of this research is to advance the integration of LLMs with autonomous web agents. It has only been in the last year that researchers have begun to utilize LLMs outside of NLP and integrate them as core capabilities in autonomy (Lu et al. (2021); Ahn et al. (2022)). In this context, LLMs are reasoning engines for sequential decision making agents interacting with environments.

The present work is the first in the research literature to embed an LLM and train it as an agent for autonomous web navigation. This requires new implementations to adapt LLM training for behavior cloning in addition to designing interfaces for integrating text generation into a perception-compute-

action cycle operating in a stateful web environment. Our implementation allows us to answer new questions regarding trade-offs among various model characteristics.

We believe these contributions expand the scope of language models and connect their unique capabilities with autonomous agents for the web. We provide a new perspective on machine learning for HTML understanding and web automation, showing that pretrained LLMs can achieve significant performance on such tasks, reducing the need for specialized architectures and training protocols. To encourage further research in this direction, we open sourced [2] model weights for agents used in the WoB environment and plan to open-source our dataset for description generation and.

## 2 RELATED WORK

**HTML Understanding** Autonomous web navigation has been a popular application for neural network models, and a variety of works propose simulated websites for training web-based agents, with application to task fulfillment (Yao et al., 2022; Gur et al., 2021; Burns et al., 2022; Mazumder & Riva, 2020; Shi et al., 2017; Liu et al., 2018) as well as information retrieval or question-answering (Adolphs et al., 2021; Nogueira & Cho, 2016). Simulated websites provide an easy way to evaluate models online, and for this reason we use the existing MiniWoB benchmark (Shi et al., 2017) for our web navigation setting. However, it is still important to have a mechanism for evaluating models on a wide variety of real-world websites. This was the key motivation for generating our own dataset for the description generation task, which is distilled and auto-labeled from CommonCrawl and is a key contribution of our paper.

Alongside these benchmarks, many works have developed models for web navigation and related subtasks (Pasupat et al., 2018; Bommasani et al., 2021; He et al., 2021; Gur et al., 2021; Humphreys et al., 2022; Liu et al., 2018; Jia et al., 2019). These works often rely on specialized neural network architectures that leverage inductive biases of HTML structure, or on preprocessing of HTML to make it easier to input to a model (Li et al. (2021a;b)). In contrast, our work takes a minimalist approach, providing HTML in text form with minimal processing and using widely-adopted transformer networks.

**LLMs and HTML** Works that explore the intersection of LLMs and HTML generally fall into two categories. The first category uses LLMs to assist web navigation (Nakano et al., 2021; Yao et al., 2022), and typically relies on a custom preprocessing to map the context and structure of a web page to natural language, thus severely restricting what HTML pages the model can parse. The second category pretrains LLMs on a large corpora of HTML text (Aghajanyan et al., 2021). However, these works typically restrict the model evaluation to standard NLP tasks, e.g., summarization and question/answering as opposed to tasks more relevant to HTML understanding and web automation. Our work can be thought of as the reverse: We keep the pretraining of LLMs unchanged and focus on the mechanisms for transferring the pretrained LLMs to HTML-relevant tasks.

## 3 BRIEF BACKGROUND ON HTML AS SEMI-STRUCTURED TEXT DATA

HTML is a markup language, used to organize web page **structure** and **content**. Consider the example HTML page in Figure 1a. This web page includes two adjacent `input` elements, one for e-mail and another for password, with their corresponding `labels` on a separate branch of the page. These `inputs` and `labels` are one of many possible *elements* that serve as HTML building blocks. Each element has a set of attributes – key and value pair – that describe the element's content, such as style and human-readable text. When rendered in a browser, these attributes will be responsible for how the element is shown and where it is positioned. In the example in Figure 1a, the first `input` has three attributes, `tag="input"`, `type="email"`, and `id="uName"`, that identify the element as an email input with an identifier ("uName") that can be accessed programmatically.

---

[2]`https://drive.google.com/corp/drive/folders/1aNXHyj-PU3hJcaofWqabRh3urmr4H_g-`

| Task | Dataset | Size | Input | Model Architecture | Output | Task Output |
|---|---|---|---|---|---|---|
| *Autonomous Web Navigation* | MiniWoB Demos (Shi et al., 2017) | 12K | Page | Enc-Dec Dec | Text | Dictionary |
| *Semantic Classification* | Annotated Shopping Webpages (Gur et al., 2021) | 28K | Snippet | All | Text | Category |
| *Description Generation* | CommonCrawl (new) | 85K | Snippet | Enc-Dec Dec | Text | Text |

Table 1: Task, dataset, and model summary. All models receive raw HTML. *Autonomous Web Navigation* receives the entire HTML, while the other tasks receive HTML snippets extracted given salient element.

## 4 CANONICAL TASKS FOR HTML UNDERSTANDING

We devise three canonical tasks to study HTML understanding capabilities of LLM-based web agents. These tasks require correctly interpreting both structure and content to varying degrees to make predictions, with autonomous navigation being the most challenging capability of the three.

***Autonomous Web Navigation***. This task evaluates how well a model navigates multi-page websites as a sequential decision-making problem (Shi et al., 2017; Liu et al., 2018). At the beginning of an episode, the agent is given a natural language instruction, e.g. *Enter the username "lyda" and the password "N22t" into the text fields and press login*. The agent applies actions to a sequence of HTML pages, where each action is of the form `function(selector, text)`. The `function` is one of *click* or *type*, `selector` is an integer pointer that uniquely identifies an element, and `text` is a text to input if the *type* functionality is activated. An episode terminates when either the page reaches a terminal state (e.g., the 'sign in' button is clicked) or the maximum number of steps is reached.

***Semantic Classification***. Many HTML understanding applications require a model that can classify HTML elements into standardized categories. For example, in automated form-filling (Diaz et al., 2013; Gur et al., 2021), it is useful to identify a 'submit button' across many websites (e.g., shopping, flight booking, utility application) with various button representations (e.g., position, color, or text). Thus, we formulate *Semantic Classification* as classifying elements into *role* categories. Take the example HTML in Figure 1a which includes two `input` elements and a submit `button`. Let's pick the first `input` as an element of interest to be classified by the system, also called a *salient element*. The system should classify this element as *username*, since it appears on a login page and it has a `label` with *Email Address* which is typically associated with the username in form-filling applications. To solve this, the system can aggregate information from multiple sources in the page – the label that says *Enter Email Address*, the `input` attributes (*type="email"* and *id="uName"*), or even the ordering of other elements in the page such as 'password' and 'sign in'.

***Description Generation***. Motivated by applications in accessibility-minded web browser control (Jorgensen & Binsted, 2005), we formulate description generation as an extractive problem where the goal is to locate the textual description of an element in the HTML and generate it as output. For instance, the description of the salient element in Figure 1a is *Enter Email Address*; when rendered, this `label` will appear above the 'email' `input` field. HTML provides a large amount of flexibility, and so in general a descriptive text that appears alongside a specific element when rendered can be very far from that element when looking at the HTML plaintext. Thus, this task evaluates a model's ability to understand the structure of HTML as it would appear to a user, despite not having access to the rendered web page directly.

## 5 DATASETS

Each of our canonical tasks requires a separate dataset, with the description generation task using a newly contributed, auto-labelled dataset based on CommonCrawl.

***Autonomous Web Navigation***. We use the 12K demonstrations included in the publicly available MiniWoB benchmark (Shi et al., 2017), which encompass 62 website applications ranging from email forwarding to social media interactions. Each demonstration is a sequence of (**instruction, HTML, action**) tuples. Every element in a MiniWoB demonstration is accompanied by a reference number unique within its respective pages. This number can be used as an element selector, making the action space unified across all tasks and time steps. For instance, the action in Figure 1a would be

*type(ref=5, "username@email.com")*, where 5 refers to the index of the input when counted from top-to-bottom. As model input, we concatenate the natural language instruction and HTML into a single text input sequence. Similarly, we treat the action as a text sequence for the model to predict.

***Semantic Classification.*** We use a dataset of 28K labelled examples, containing 66 different categories, of the form **(HTML, element, category)**, previously used in the context of environment generation (Gur et al., 2021). The dataset consists of HTMLs from real-world shopping websites and categories relevant to form-filling during payment and checkout on these websites.

***Description Generation.*** For this task, we derive a dataset from CommonCrawl.[3] CommonCrawl does not include renderings or annotations that would reveal what text in the HTML is associated with which elements. Instead, we infer descriptions of various elements by exploiting a special attribute in the HTML schema known as `for`. As an example in Figure 1a, the first `label` in the HTML has a `for` attribute with value *uName*, which is the `id` of the element described by `label`; in this case, the `id` is that of the first `input` in the page. This annotation does not affect the rendering of the page and is typically used for accessibility purposes. We utilize the information given by these `for` attributes to create a large-scale dataset to study description generation. A small sample is available in the supplemental material, while the entire dataset will be available upon publication.

Specifically, we collected 100 WARC (from April 2019) files from the CommonCrawl project and extracted all HTML `label`s that have a `for` attribute. Removing non-Unicode and alphanumeric text in HTML `label`s results in a 400K example datset. We balance the distribution of labels, effectively downsampling the dataset to $85K$ samples. Each example is represented as **(HTML, element, description)**, where **HTML** is the HTML plaintext of the page, **element** is the element whose `id` attribute matches that appearing in the `label`'s `for` attribute, and **description** is the text inside the `label` element (see example in Figure 1a). More details of the dataset can be found in Appendix A.1.

# 6   PRE-PROCESSING

In treating HTML as token sequences, we minimize any HTML tree pre-processing prior to model input. We thus provide HTML as raw text (i.e., sequences of text tokens) and only apply a snippet extraction pre-processing for pages which are too large to fit into the typical LLMs context windows.

**Snippet Extraction.** Real HTML pages can grow extremely large, reaching thousands of elements, far beyond the context window of the largest LLM that we studied (1920 tokens in PaLM (Chowdhery et al., 2022)). LLMs typically truncate such long sequences, which can be detrimental to HTML understanding as HTMLs are not linearly structured. We take an element-centric approach and extract HTML snippets (a small portion of HTML code) surrounding a salient element (Figure 5). A simple heuristic, which controls the tree's width and depth, guides the process: Start with a salient element and traverse its ancestors in the HTML tree until a stopping condition is satisfied. As we traverse up, we estimate the height of the tree and the increased number of descendants of the new root. We stop when either metric violates a pre-defined limit and take the resulting sub-tree as the snippet. We mark the salient element using a special attribute, called *target*, to distinguish it from other elements. We perform the snippet extraction for the semantic classification and description generation datasets, and keep the full HTML pages in MiniWoB because these pages are typically much smaller than real-world HTML.

**HTML un-Parsing.** We provide the models with the unparsed plaintext HTML in the form of a sequence of tokens. This canonical representation does not require specific model architectures such as hierarchical networks (Liu et al., 2018; Gur et al., 2021) and can be fed into any LLM. We transform all datasets by converting every HTML page or snippet into a sequence. For MiniWoB, we additionally concatenate (action history, instruction, HTML) tuples into a single sequence.

---

[3]`http://commoncrawl.org`

## 7 MODEL TRAINING

We study a variety of transformer-based LLMs (Vaswani et al., 2017) with different sizes and architectures for HTML understanding tasks (Table 1). In the rest of the text, we prefix models fine-tuned for *Autonomous Web Navigation*, *Description Generation*, and *Semantic Classification* with WebN-, WebD-, and WebC-, respectively. For instance, WebD–T5-3B is the three billion parameter T5 model (Raffel et al., 2020) fine-tuned for the *Description Generation* task. The rest of this section elaborates on training details.

**Encoder-Decoder and Decoder-only Models.** We train encoder-decoder models, i.e., T5 (Raffel et al., 2020), and decoder-only models, i.e., LaMDA (Thoppilan et al., 2022) and PaLM (Chowdhery et al., 2022), with text input and text output (Figure 1b). Inputs are raw HTML pages or snippet texts; similarly, outputs are categories, natural language descriptions, or actions represented as text. Namely, for *Semantic Classification* we use the textual representation of categories, similar to previous classification problems in NLP (Raffel et al., 2020). For *Autonomous Web Navigation*, actions are converted into text by first converting them into key and value pairs and then concatenating the pairs.

Many websites in MiniWoB require multiple interactions, such as *click-button-sequence* or *click-checkboxes*, where each interaction might cause a subtle change in the website state. For instance, after clicking on a checkbox in the *click-checkboxes* website, its value flips from positive to negative or the other way around, which is not always reflected in LLMs' predictions and leads to action repetitions. We solve this issue by augmenting tuples in the dataset with a sequence of past actions, **(action history, instruction, HTML, action)**, and allowing LLMs to learn from past experience.

**Encoder-only Models.** We train encoder-only models, i.e., BERT (Devlin et al., 2018), with text input and categorical output. We keep semantic categories as discrete one-hot classes. To train encoder-only models, we add a new classification layer after the final encoder layer to produce a distribution over semantic categories. In addition to the typical BERT models, we study Mobile-BERT (Sun et al., 2020), distilled from BERT-large with inverted bottlenecks, and Albert-XL (Lan et al., 2020), with parameter sharing and embedding split.

## 8 RESULTS

We now present the results of fine-tuned LLMs for HTML understanding. We compare the models' performance with the existing baselines where possible (autonomous web navigation) and against other LLM architectures and training regimes (all tasks). Sections 8.1, 8.2, and 8.3 evaluate task-specific performance, while Section 8.4 assesses the performance across all the tasks.

**Metrics:** For autonomous web navigation we evaluate models' *Success Rate*, which is averaged over 100 episodes per task. For the other tasks, we use *Accuracy* to measure exact match between prediction and ground truth. In the description generation task, we additionally provide evaluations using alternative 'soft' text evaluation metrics, *BLEU* and *ROUGE-1*, measuring the similarity between predicted and ground truth text.

### 8.1 AUTONOMOUS WEB NAVIGATION RESULTS

For *Autonomous Web Navigation* we fine-tune two WebN- encoder-decoder architectures (WebN-T5-large and WebN-T5-3B) on 12k demonstrations from human-annotated real websites. We evaluate the models on MiniWob (Liu et al., 2018) benchmark, and compare with specialized architectures trained using supervised learning (SL) on 2.4 million human expert demonstrations *CC-Net (SL)* (Humphreys et al., 2022), and two RL models bootstrapped with SL, CC-Net (SL) (CC-Net (SL & RL) (Humphreys et al., 2022), and WGE (SL & RL) (Liu et al., 2018)). Additionally, we compare with the decoder-only architecture (WebN-Lambda-1B) and perform an ablation study on the impact of including the action history in the input.

**Comparison to SoTA.** Since previous works report success on only a subset of websites in Mini-WoB, we evaluate on 48 out of 62 websites that are common across all models. Table 8 in the Appendix reports fine-grained results while Figure 2a presents results averaged over all websites. Compared to CC-Net (SL) which is trained on all 2.4M demonstrations, WebN-T5-3B improves the

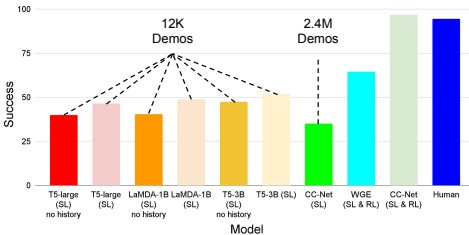

(a) Baseline comparison.

| Model Name | Success (%) | Model Size |
|---|---|---|
| T5-large | 18.1 | 800M |
| LaMDA-1B | 15.6 | 1B |
| T5-3B | 11.1 | 3B |
| WebN-T5-large | 46.4 | 800M |
| WebN-LaMDA-1B | 48.8 | 1B |
| WebN-T5-3B | 51.8 | 3B |

(b) Pre-training effect.

Figure 2: a) WebN–T5* performance compared to the previous SOTA models on MiniWoB benchmark. WebN-T5-3B improves the task success 16% while using 192 times less data, compared to the best supervised learning (SL) model, CC-Net (SL). LLMs performance is only surpassed by works utilizing RL, requiring orders of magnitude more online experience interaction with websites. b) LLMs with and without pretraining on *Autonomous Web Navigation* task. Those with pretraining (denoted by the 'WebN-' prefix) show a 2.5-4.5x performance improvement.

| Model Name | Test (%) | Dev (%) | Model Size | Code in training Corpus |
|---|---|---|---|---|
| WebC-MobileBERT | 78.1 | 77.7 | 24.6 M | |
| WebC-Albert-XL | 83.5 | 83.1 | 58.9 M | |
| WebC-BERT-smallest | 84.4 | 83.6 | 38.7 M | |
| WebC-BERT-small | 84.4 | 85.2 | 52.8 M | |
| WebC-BERT-medium | 85.2 | 84.5 | 67 M | 0% |
| WebC-BERT-base | 83.9 | 84.8 | 109.5 M | |
| WebC-BERT-large | 84.1 | 85.8 | 335.2 M | |
| WebC-T5-base | 86.8 | 89.9 | 250 M | |
| WebC-T5-large | 87.0 | 89.3 | 800 M | |
| WebC-T5-3B | 87.7 | 90.3 | 3 B | |
| WebC-LaMDA-1B | 87.4 | 87.1 | 1 B | 12.5% Code |
| WebC-PaLM-8B | 86.6 | 89.9 | 8 B | 5% Code (0.875% HTML) |
| WebC-PaLM-62B | **88.7** | **90.5** | 62 B | 5% Code (0.875% HTML) |
| T5-large | 76.4 | 75.2 | 800 M | |
| T5-3B | 77.2 | 73.8 | 3 B | 0% |
| PaLM-8B | 73.3 | 70.1 | 8 B | |

Table 2: LLMs performance on the *Semantic Classification* task. Fine-tuning off-the-shelf pretrained LLMs (model names with prefix 'Web*') helps LLMs transfer better compared to training the same architecture from scratch on the HTML dataset (model names without prefix 'Web*'), improving the accuracy of PaLM-8B more than 12%. While WebC-PaLM-62B clearly performed better than all other models, we found WebC-T5-large to be competitive with much larger models such as WebC-LaMDA-1B or WebC-PaLM-8B.

success 16% while only training on 12K publicly-available demonstrations, yielding over 192x improvement in sample-efficiency. We find that all choices of LLMs outperform previous SL models. Notably, WebN-T5-3B significantly improves on websites requiring multiple-action sequences such as *click_checkboxes* or websites requiring entering text such as *login_user* (Table 8). We observe that the performance of LLMs is only surpassed by previous works utilizing RL, which require orders of magnitude more online experience interaction. Extending our fine-tuned LLMs to an RL setting is a promising avenue for future work.

**Action history ablation.** Across all LLMs we consistently observe a decrease in success, on average 6.4%, when past actions are excluded from the inputs (Figure 2a). Action history helps with websites that require entering multiple texts, as well as understanding minor changes that could be difficult to detect (e.g. *click_checkboxes* and *multi_layout*). *multi_layout* requires entering 3 different texts in the website where the layout is randomized at each episode, yet, surprisingly, even the (relatively smaller) WebN-T5-large model without action history outperforms the CC-Net (SL) model; illustrating that incorporating action history is not the only contributing factor for the better success.

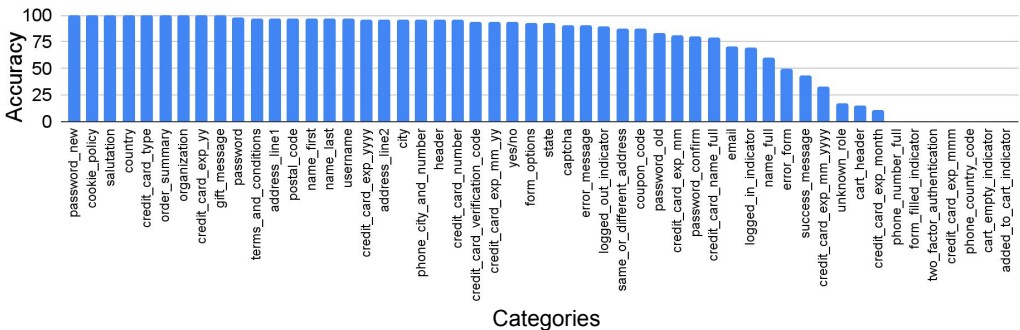

Figure 3: Accuracy per classification category of the WebC-T5-3B model on the development dataset.

| New descendants (%) | Height | Test (%) | Dev (%) |
|---|---|---|---|
| 25 | 3 | 87.7 | 90.3 |
| 25 | 4 | 88.6 | 89.2 |
| 50 | 3 | 88.4 | 90.0 |
| 50 | 4 | 89.3 | 89.2 |
| 300 | 5 | 87.8 | 88.8 |
| 500 | 7 | 75.8 | 74.5 |

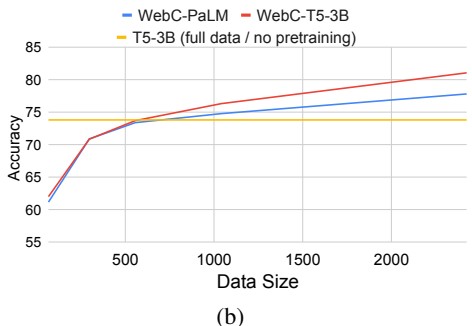

(a)                                                                 (b)

Figure 4: a) Effect of snippet extraction parameters on WebC-T5-3B. Increases above 50% in new descendants and height of 4. Large increases in both parameters lead to large snippets and decrease in accuracy. b) Accuracy over training data size. Using only 1000 labeled examples (4.4% of all training dataset), WebC-T5-3B outperforms T5-3B (full data without pretraining) which is trained on *all* available labeled data (approximately 30k examples), and outperforms WebC-PaLM-8B which is an order of magnitude larger.

## 8.2 SEMANTIC CLASSIFICATION TASK RESULTS

To evaluate the *Semantic Classification* task, we compare the T5 encoder-decoder architecture's three size variants (WebC-T5-base, WebC-T5-large, and WebC-T5-3B) fine-tuned on 22K real, human-labeled training websites. We compare with a fine-tuned encoder only architectures (WebC-*BERT*), three fine-tuned decoder-only architectures (WebC-LaMDA and PaLM), and both encoder-decoder and decoder-only models trained on human labeled websites from scratch. Results are presented in Table-2, where we find that all WebC-LLMs perform well and significantly better than the same architectures without pretraining.

**Accuracy per category.** In Figure 3, we present accuracy distribution of the WebC-T5-3B model on the development dataset. The fine-tuned encoder-decoder model performs strongly on a majority of the categories (Figure 3), even on those with very few samples. For instance, the model is 100% accurate on *password_new* which has only 56 training examples, because the class is unambiguous. On the other hand, unsurprisingly, the performance drops when the category is ambiguous, such as in the *email* category which is frequently mistaken as *username*.

**Snippet generation ablation.** Two hyper-parameters govern snippet generation: percentage of new descendants and height of the new root. While small variations of both parameters do not change the performance, increasing both degrades the performance significantly (Table 4a). With new descendants up to 500% and height up to 7, the performance drops by more than 15%. Note that snippet generation returns the full-page HTML when both parameters increase indefinitely.

**Data size impact.** When varying the fine-tuning training data sizes (1, 5, 10, 20, or 50 samples per class) in Figure 4b, WebC-T5-3B slightly outperforms WebC-PaLM-8B which is an order of magnitude larger. Compared to T5-3B that is trained on all available HTML data without pretraining, WebC-T5-3B achieves better performance while using only 3.4% of labeled data (1000 samples),

| Model Name | Test | | | Dev | | |
| --- | --- | --- | --- | --- | --- | --- |
| | Accuracy(%) | BLEU | ROUGE-1 | Accuracy(%) | BLEU | ROUGE-1 |
| WebD-T5-large | 83.2 | 90.2 | 90.5 | 84.3 | 91.7 | 91.5 |
| WebD-LaMDA-1B | 83.3 | 87.5 | 90.2 | 84.3 | 88.6 | 91.2 |
| WebD-T5-3B | 84 | 90.8 | 90.9 | 85.2 | 92.1 | 91.9 |
| Closest Description | 57.4 | 24.4 | 59.2 | 60.8 | 23.9 | 62.1 |

Table 3: Description generation accuracy of LLMs.

thus highlighting the benefit of using standard off-the-shelf pretrained LLMs for HTML understanding.

## 8.3 DESCRIPTION GENERATION TASK RESULTS

For *Description Generation* we split the CommonCrawl dataset based on URL top-level domains to test LLMs' capabilities to generalize to unseen HTML. We fine-tune encoder-decoder architectures (WebD–T5*) and decoder-only models (WebD–LaMDA*), with results presented in Table 3. We also evaluate a strong heuristic baseline which simply finds the description closest to the salient element in the HTML text (Closest Description).

**Accuracy and Similarity Performance** We show results of our evaluations in Table 3. All models achieve high scores across all metrics, achieving $\approx 84\%$ on the accuracy in terms of exact match and a higher non-exact match score based on BLEU and ROUGE-1 ($\approx 91\%$). This difference indicates that the models are capable of locating the descriptions, but not always generating the exact output.

## 8.4 HTML UNDERSTANDING LLMS PERFORMANCE ANALYSIS ACROSS TASKS

We now analyze our results in aggregate to derive our main conclusions.

### 8.4.1 PRETRAINING EFFECT: PRETRAINING ON LARGE TEXT CORPORA MATTERS

Fine-tuned pretrained LLMs outperform LLMs trained on HTML-only data, improving the performance by more than 34.1% on the *Autonomous Web Navigation* (Table 2b), and 10% to 12.7% on the *Semantic Classification* task (Table 2).

Since *Autonomous Web Navigation* is the most difficult task, the improved performance is an encouraging evidence of the value of LLMs in HTML understanding tasks. Specifically, we observe that LLMs without pretraining are comparable to fine-tuned pretrained models only on websites that require simple text matching. In contrast, for websites such as *click_checkboxes*, text matching is harder and we find that pretraining is key to good performance. We also found that without pretraining, model outputs were frequently in an incorrect format such as invalid dictionaries or invalid refs with non-integer values. This suggests that the large corpora used for pretraining helps models to learn general HTML structure.

### 8.4.2 ARCHITECTURE EFFECT: T5-BASED MODELS PERFORM BEST ACROSS ALL TASKS

Encoder-decoder T5 based models perform better across all three tasks. On the *Autonomous Web Navigation* task, encoder-decoder (WebN-T5) architectures are better or comparable to WebN-LaMDA-1B (Figure 2a). On the *Semantic Classification*, the smallest encoder-decoder model (WebC-T5-base) performs comparably to much larger decoder-only models (WebC-LaMDA-1B or WebC-PaLM-8B) and the largest encoder-only model (WebC-BERT-large) which has 85M more parameters (Table 2). We also observe that decoder-only PaLM-8B performs worse than much-smaller encoder-decoder T5-large when trained only on HTML data. Finally, on the *Description Generation* encoder-decoder architecture has higher BLEU score.

One possible explanation for the strong performance of T5-based moels is the encoder-decoder architecture of these models. Namely, T5 models utilize an encoder with a bidirectional attention mechanism, not present in the LaMDA and PaLM decoders. The bidirectional attention mechanism can process HTML pages from both ends, potentially overcoming the loss of information when tree-structured HTML pages are converted into a fixed linear text sequences.

### 8.4.3 Model Size Effect: Size (Sub-linearly) Matters

Across the tasks it appears that the architecture plays an important role in the model performance. Model size and performance are also positively correlated, although they reach diminishing returns. For instance, the model performance is roughly $O(\log \log n)$ with respect to model size on *Semantic Classification* (Figure 4b in Appendix). On the *Autonomous Web Navigation* task, performance grows slowly with the model size (Table 8), while on the *Description Generation* it plateaus (Table 3).

### 8.5 Discussion

**Bi-directional attention vs training corpora:** Pretraining on large corpora matters, yielding ≤4.5x performance improvements. Larger models tend to be better and we credit the bidirectional attention for T5's best overall performance across the tasks. PaLM and LaMDA include HTML and other code in their pretraining corpora, while BERT and T5 architectures did not, showing that pretraining on HTML is not necessary for strong performance when fine-tuned for HTML understanding. This strengthens the hypothesis behind the role of the bidirectional attention, and opens up the possibility to further improve the performance of T5 architectures by pretraining them on corpora with HTML.

**Practical impact on labeling:** When available, the pretrained LLMs need very little new expert data (200x and 30x reduction on the web navigation and classification tasks, respectively). This has a big potential impact on practical applications, reducing the data collection time and cost by orders of magnitude.

**Bigger is not always better:** When choosing the model size, the expected performance gains (sub-linear at best and asymptotic at worst) should be considered alongside the model's training and inference time and cost. For instance, on the classification task, the largest model WebC-PaLM-62B takes several days to fine-tune, and evaluates at 30 Hz, while WebC-T5-large fine-tunes in several hours and evaluates at 700 Hz – an order of magnitude more expensive for a single percent uplift in accuracy. BERT models on the other hand train in minutes. If the application does not require high precision, these might be a good choice.

**Context window is a bottleneck:** The major bottleneck for the HTML understanding tasks seems to be the context window length that the current LLMs support, even with models that accept 1000+ tokens. It remains prohibitive to evaluate web navigation tasks on real websites that are orders of magnitude larger than pages in MiniWob. Similarly, we observed that increasing the snippet size leads to major performance degradation. This makes HTML understanding an interesting benchmark for future LLM development. For instance, new methods may need to be developed to compress the state representation of web content for use in LLM context windows.

## 9 Conclusion

We presented canonical tasks and fine-tuned LLMs for HTML understanding. The comprehensive evaluations and analyses over a range of architectures, dataset sizes, and baselines yields practical findings and highlights current limitations of these models. We find that a) pretraining is critical for the performance and can reduce labeled data requirements, improving sample efficiency up to 200x; b) model architecture is the second-most important factor, and T5 models with bidirectional attention and encoder-decoder architecture perform the best across the board; c) given a choice, model size should be evaluated in the context of the model's training and inference performance, as the model size sub-linearly correlates with its performance. Finally, the proposed HTML understanding tasks highlight the relatively short context window that limits current LLMs, suggesting possibilities for future research that incorporate or eliminate this constraint.

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

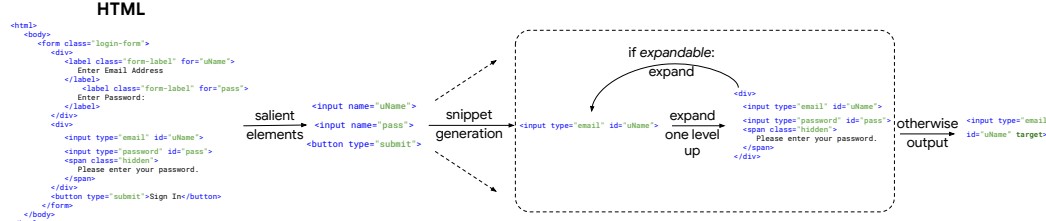

Figure 5: High-level overview of our pre-processing pipeline for generating snippets from a full HTML web-page. Given the page, we detect salient elements and for each one of them we extract snippets by recursively moving up in the HTML tree until a validation heuristic fails.

Thomas Wolf, Lysandre Debut, Victor Sanh, Julien Chaumond, Clement Delangue, Anthony Moi, Pierric Cistac, Tim Rault, Rémi Louf, Morgan Funtowicz, and Jamie Brew. Huggingface's transformers: State-of-the-art natural language processing. *CoRR*, abs/1910.03771, 2019. URL http://arxiv.org/abs/1910.03771.

Linting Xue, Noah Constant, Adam Roberts, Mihir Kale, Rami Al-Rfou, Aditya Siddhant, Aditya Barua, and Colin Raffel. mt5: A massively multilingual pre-trained text-to-text transformer. *arXiv preprint arXiv:2010.11934*, 2020.

Shunyu Yao, Howard Chen, John Yang, and Karthik Narasimhan. Webshop: Towards scalable real-world web interaction with grounded language agents. *arXiv preprint arXiv:2207.01206*, 2022.

## A    APPENDIX

### A.1    DATASET DETAIL

Examining the description distribution, we found the original $400K$ dataset to be very skewed; only 20 descriptions (such as *Email* and *Password*) were covering 50% of the dataset. We sub-sampled the dataset so that each unique description has at most 10 data points. We also found that `for` attributes are almost always defined for HTML `label`s. This could cause a model to overfit and just find the `label` element in the HTML and ignore everything else. To avoid this sort of 'cheating' we replace the tags of HTML `label`s by randomly sampling from {`div`, `span`, `a`, `label`}. These tags are also frequently used to inject text in HTML but they are very rarely used with `for` attributes. Finally, we removed examples where there are only a single text in the HTML since models can trivially generate descriptions by finding the only text in the HTML, which biases model weights and evaluation metrics. After this final step, we have a total of $85K$ labeled examples.

#### A.1.1    SNIPPET GENERATION

In Figure 5, we give a high-level overview of our snippet generation procedure.

### A.2    ADDITIONAL RESULTS

#### A.2.1    SEMANTIC CLASSIFICATION

**Error Analysis.**    We manually examined 50 errors of T5-3B model over the development set (Table 4) and assigned them into one of the 9 error types that we devised. We found that 32% of the errors are due to lack of information in the HTML snippets, which is mainly the result of lost information during snippet extraction process. Annotation errors or email/username ambiguity make up 30% of the errors. These can't be improved without revising the annotated data or adding extra information to resolve the ambiguity. We also found that the model sometimes picks a more general category, or a nearby text misleads the model; the latter usually happens when the HTML snippet is long where majority of the elements are noise.

| Error Type | Percentage of Examples |
|---|---|
| Not enough information in the HTML snippet | 30 |
| Incorrect annotation (ex: "unknown_role" instead of "organization") | 12 |
| Annotation tool translates user selection incorrectly | 8 |
| Email/Username ambiguity | 10 |
| More general category (ex: "header" instead of "cart_header") | 8 |
| Immediate neighboring text misleads | 8 |
| Incorrect date formatting (ex: "mm" instead of "mmm") | 4 |
| No information in the HTML snippet | 2 |
| Others | 18 |

Table 4: Types of errors over 50 manually examined examples. 32% of errors are due to lack of information in HTML snippets, 30% of errors are related to annotations or can't be improved due to ambiguity (email/username), and the remaining errors are incorrect predictions by the model.

**Few-Shot Prompting** In Table 5, we present few-shot prompting performance of a 540B PaLM model. We probe the model using a prompt template `<html> Role:  <category>` with 1 example per category and generate categories using greedy-decoding. In our preliminary experiments, we found that few-shot prompting achieves only 45.6 accuracy, much lower than a model fine-tuned on the same data (Figure 6). We found two common problems – the model is not able to canonicalize predictions into categories and many of the examples are dropped due to context length.

We developed post-processing methods to alleviate the canonicalization problem and pre-processing methods to reduce lengths of examples. Adding a dictionary-based mapping on predictions – a manually curated paraphrase dictionary – improves the performance to 52.1. We also tried rewriting predictions by changing the order of tokens around "_" such as *name_first* to *first_name* which further improved the performance to 57.9. Finally, we cleaned examples in the prompt by removing certain elements such as *"svg", "path", "img"*, and *"iframe"* and also removing *class* attribute from every element; this pre-processing step gives 64.2.

| Model Name | Test | Dev |
|---|---|---|
| PaLM-540B | 64.2 | 60.3 |
| - w/o Example Cleaning | 57.9 | 57.2 |
| - w/o Category Rewriting | 52.1 | 50.7 |
| - w/o Dictionary Mapping | 45.6 | 45.1 |

Table 5: Few-shot prompting performance with different pre- and post-processing steps.

Figure 6: Performance comparison w.r.t. increasing model size. As the model size increases, we observe an increase in overall accuracy with PaLM-62B model achieving the highest accuracy while being 7x larger than PaLM-8B.

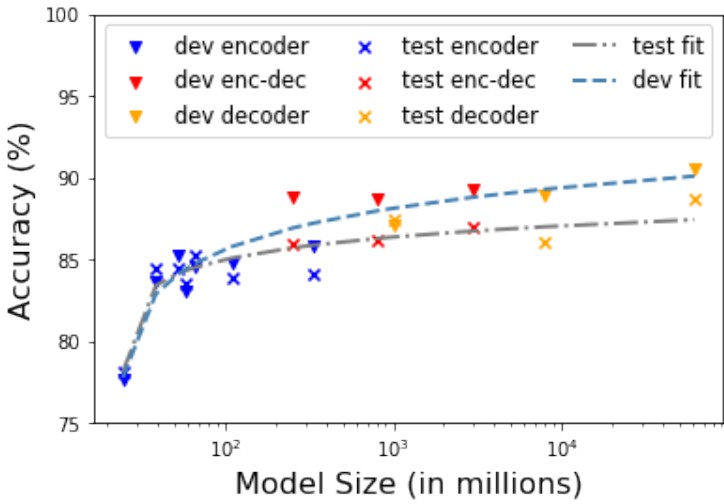

### A.3 Sample Episodes from MiniWoB

See Table 6 for an example episode of web navigation inferred by a fine-tuned LLM.

### A.4 Detailed MiniWoB Results

See Table 7 for detailed performance of various models on MiniWob.

### A.5 Resource Requirements

See Table 8.

### A.6 Structure Dependence Ablation Study

We conducted an ablation study to examine the sensitivity of model performance to preserving structural information. To do so, we evaluate the model's performance on HTML input with critical structure components removed. We kept the order of elements and their attributes fixed while corrupting the nesting structure by removing closing tags.

Removing closing tags corresponds to a valid traversal (BFS) and keeps the order of elements the same as the text based input.

As a simple example:

```
<div id="form"><div><input id="username"></div></div>
```

would be converted into:

```
<div id="form"><div><input id="username">
```

We evaluated the trained WebN-T5-3B model on the same set of synthetic websites from the MiniWoB benchmark with this aspect of structure removed from the HTML pages. WebN-T5-3B achieves a 45.4% success rate, 6% lower than before, suggesting that WebN-T5-3B is at least partially dependent on the DOM topology.

### A.7 Task-specific Models

An alternative to LLMs is to adapt bespoke task-specific architectures tailored towards processing of structured documents and HTML (Li et al. (2021b;a)).

StructuralLM (Li et al. (2021a)) is an approach specifically tailored for document understanding (i.e., combinations of images and text), and thus makes several simplifying assumptions for its model that limit its applicability to HTML understanding (i.e., trees of elements with a richer structure and functionality). It is trained only on the textual content of a document - the markup information is ignored. For example, any input field or dropdown in a document would be missing from the model inputs. All of the tasks we study require knowledge of this information. For example, in autonomous navigation the model needs to interact with input elements (e.g. text, checkboxes, dropdowns) such as username and password in the login-user task in MiniWoB. Typically, a "type" action with a reference to an element and a text argument is generated by the model. Without knowing which input elements are available in the page, it is impossible to generate a reference to any input element.

While MarkupLM (Li et al. (2021b)) is better tailored for understanding HTML pages, it has similar drawbacks as StructuralLM in that it focuses solely on text and structure of text while ignoring everything else in the markup. To illustrate our point better, we used the open source implementation of MarkupLM from the HuggingFace library (Wolf et al. (2019)) to process the sample HTML snippet in Figure-1(b). The MarkupLM ignores all input elements, both username and password, and generates *Email AddressEnter Password:Please enter your password.* which is the text input to the MarkupLM Transformer. Classifying this text as username or password is not possible without the additional context on which input element is the salient element (in this context it is the username). See below for the code to reproduce our result.

```
from transformers import MarkupLMProcessor
processor = MarkupLMProcessor.from_pretrained(f"microsoft/markuplm-base")
snippet = '''<div><label class="form-label" for="uName">Email Address
</label><label class="form-label" for="pass">Enter Password:
</label></div><div><input type="email" id="uName" target><input
type="password" id="pass">Please enter your password.
</div>'''
encoding = processor(snippet)
print(processor.batch_decode(encoding["input_ids"]))
```

MarkupLM is also evaluated on NLP-like tasks such as QA or entity classification where understanding page content is paramount, whereas we focus on HTML understanding tasks such as autonomous navigation where both content and the page's layout structure need to be understood.

We perform a quantitative evaluation of MarkupLM on our tasks to understand how significant these limitations are. We fine-tune the MarkupLM-base model on the semantic classification task, using the same setup as other WebC models but with the suggested hyperparameters from (Li et al. (2021b)). We use the MarkupLM implementation from the HuggingFace library (Wolf et al. (2019)). On development and test sets, MarkupLM-base achieves 65% and 66% accuracy, respectively. These results are more than 16% lower compared to similar size WebC-BERT-base results that we report in our work. This suggests that although domain specific models may be suitable for processing HTML for NLP tasks, the generality, flexibility, and sample efficiency LLMs provide advantages for autonomous navigation tasks.

Table 6: A sample web page and corresponding episode using the T5-3B model. At each time step, previous actions, instruction, and HTML are concatenated into a single HTML text. Note that at the beginning of episode, there is no past actions and we simply concatenate instruction and HTML. Action is generated as a sequence of tokens which is later parsed into a dictionary. The *ref* in the action points to an element that has a *ref* attribute with the same value. For instance, at the beginning of episode, *ref: 6* corresponds to an input with *ref=6*. At the end of the episode, the model clicks on the submit button and the episode terminates.

Web page

| HTML Text | Action Text |
|---|---|
| Select qw, mMpY, YVYMQVa, DJfG, lI3Z and click Submit. \<body ref="1">\<div id="wrap" ref="2">\<div id="area" ref="3">\<div id="boxes" ref="4">\<label ref="5">\<input type="checkbox" id="ch0" ref="6" value="False">\</input>\<t class="TEXT\_CLASS" ref="None">qw\</t>\</label>\<label ref="7">\<input type="checkbox" id="ch1" ref="8" value="False">\</input>\<t class="TEXT\_CLASS" ref="None">hHfDn8\</t>\</label>\<label ref="9">\<input type="checkbox" id="ch2" ref="10" value="False">\</input>\<t class="TEXT\_CLASS" ref="None">mMpY\</t>\</label>\<label ref="11">\<input type="checkbox" id="ch3" ref="12" value="False">\</input>\<t class="TEXT\_CLASS" ref="None">YVYMQVa\</t>\</label>\<label ref="13">\<input type="checkbox" id="ch4" ref="14" value="False">\</input>\<t class="TEXT\_CLASS" ref="None">DJfG\</t>\</label>\<label ref="15">\<input type="checkbox" id="ch5" ref="16" value="False">\</input>\<t class="TEXT\_CLASS" ref="None">lI3Z\</t>\</label>\</div>\<button id="subbtn" class="secondary-action" ref="17">Submit\</button>\</div>\</div>\</body> | {action: click, ref: 6} |
| {action: click, ref: 6} Select qw, mMpY, YVYMQVa, DJfG, lI3Z and click Submit. \<body ref="1">\<div id="wrap" ref="2">\<div id="area" ref="3">\<div id="boxes" ref="4">\<label ref="5">\<input type="checkbox" id="ch0" ref="6" value="True">\</input>\<t class="TEXT\_CLASS" ref="None">qw\</t>\</label>\<label ref="7">\<input type="checkbox" id="ch1" ref="8" value="False">\</input>\<t class="TEXT\_CLASS" ref="None">hHfDn8\</t>\</label>\<label ref="9">\<input type="checkbox" id="ch2" ref="10" value="False">\</input>\<t class="TEXT\_CLASS" ref="None">mMpY\</t>\</label>\<label ref="11">\<input type="checkbox" id="ch3" ref="12" value="False">\</input>\<t class="TEXT\_CLASS" ref="None">YVYMQVa\</t>\</label>\<label ref="13">\<input type="checkbox" id="ch4" ref="14" value="False">\</input>\<t class="TEXT\_CLASS" ref="None">DJfG\</t>\</label>\<label ref="15">\<input type="checkbox" id="ch5" ref="16" value="False">\</input>\<t class="TEXT\_CLASS" ref="None">lI3Z\</t>\</label>\</div>\<button id="subbtn" class="secondary-action" ref="17">Submit\</button>\</div>\</div>\</body> | {action: click, ref: 10} |
| {action: click, ref: 6},{action: click, ref: 10} Select qw, mMpY, YVYMQVa, DJfG, lI3Z and click Submit. \<body ref="1">\<div id="wrap" ref="2">\<div id="area" ref="3">\<div id="boxes" ref="4">\<label ref="5">\<input type="checkbox" id="ch0" ref="6" value="True">\</input>\<t class="TEXT\_CLASS" ref="None">qw\</t>\</label>\<label ref="7">\<input type="checkbox" id="ch1" ref="8" value="False">\</input>\<t class="TEXT\_CLASS" ref="None">hHfDn8\</t>\</label>\<label ref="9">\<input type="checkbox" id="ch2" ref="10" value="True">\</input>\<t class="TEXT\_CLASS" ref="None">mMpY\</t>\</label>\<label ref="11">\<input type="checkbox" id="ch3" ref="12" value="False">\</input>\<t class="TEXT\_CLASS" ref="None">YVYMQVa\</t>\</label>\<label ref="13">\<input type="checkbox" id="ch4" ref="14" value="False">\</input>\<t class="TEXT\_CLASS" ref="None">DJfG\</t>\</label>\<label ref="15">\<input type="checkbox" id="ch5" ref="16" value="False">\</input>\<t class="TEXT\_CLASS" ref="None">lI3Z\</t>\</label>\</div>\<button id="subbtn" class="secondary-action" ref="17">Submit\</button>\</div>\</div>\</body> | {action: click, ref: 12} |
| {action: click, ref: 6},{action: click, ref: 10},{action: click, ref: 12} Select qw, mMpY, YVYMQVa, DJfG, lI3Z and click Submit. \<body ref="1">\<div id="wrap" ref="2">\<div id="area" ref="3">\<div id="boxes" ref="4">\<label ref="5">\<input type="checkbox" id="ch0" ref="6" value="True">\</input>\<t class="TEXT\_CLASS" ref="None">qw\</t>\</label>\<label ref="7">\<input type="checkbox" id="ch1" ref="8" value="False">\</input>\<t class="TEXT\_CLASS" ref="None">hHfDn8\</t>\</label>\<label ref="9">\<input type="checkbox" id="ch2" ref="10" value="True">\</input>\<t class="TEXT\_CLASS" ref="None">mMpY\</t>\</label>\<label ref="11">\<input type="checkbox" id="ch3" ref="12" value="True">\</input>\<t class="TEXT\_CLASS" ref="None">YVYMQVa\</t>\</label>\<label ref="13">\<input type="checkbox" id="ch4" ref="14" value="False">\</input>\<t class="TEXT\_CLASS" ref="None">DJfG\</t>\</label>\<label ref="15">\<input type="checkbox" id="ch5" ref="16" value="False">\</input>\<t class="TEXT\_CLASS" ref="None">lI3Z\</t>\</label>\</div>\<button id="subbtn" class="secondary-action" ref="17">Submit\</button>\</div>\</div>\</body> | {action: click, ref: 14} |

```
{action: click, ref: 6},{action: click, ref: 10},{action: click, ref: 12},
{action: click, ref: 14} Select qw, mMpY, YVYMQVa, DJfG, lI3Z and click Submit.
<body ref="1"><div id="wrap" ref="2"><div id="area" ref="3"><div id="boxes"
ref="4"><label ref="5"><input type="checkbox" id="ch0" ref="6" value="True">
</input><t class="TEXT\_CLASS" ref="None">qw</t></label><label ref="7"><input
type="checkbox" id="ch1" ref="8" value="False"></input><t class="TEXT\_CLASS"
ref="None">hHfDn8</t></label><label ref="9"><input type="checkbox" id="ch2"
ref="10" value="True"></input><t class="TEXT\_CLASS" ref="None">mMpY</t></label>
<label ref="11"><input type="checkbox" id="ch3" ref="12" value="True"></input>
<t class="TEXT\_CLASS" ref="None">YVYMQVa</t></label><label ref="13"><input
type="checkbox" id="ch4" ref="14" value="True"></input><t class="TEXT\_CLASS"
ref="None">DJfG</t></label><label ref="15"><input type="checkbox" id="ch5"
ref="16" value="False"></input><t class="TEXT\_CLASS" ref="None">lI3Z</t>
</label></div><button id="subbtn" class="secondary-action" ref="17">Submit
</button></div></div></body>
```
{action: click, ref: 16}

```
{action: click, ref: 6},{action: click, ref: 10},{action: click, ref: 12},
{action: click, ref: 14},{action: click, ref: 16} Select qw, mMpY, YVYMQVa,
DJfG, lI3Z and click Submit. <body ref="1"><div id="wrap" ref="2"><div id="area"
ref="3"><div id="boxes" ref="4"><label ref="5"><input type="checkbox" id="ch0"
ref="6" value="True"></input><t class="TEXT\_CLASS" ref="None">qw</t></label>
<label ref="7"><input type="checkbox" id="ch1" ref="8" value="False"></input><t
class="TEXT\_CLASS" ref="None">hHfDn8</t></label><label ref="9"><input
type="checkbox" id="ch2" ref="10" value="True"></input><t class="TEXT\_CLASS"
ref="None">mMpY</t></label><label ref="11"><input type="checkbox" id="ch3"
ref="12" value="True"></input><t class="TEXT\_CLASS" ref="None">YVYMQVa</t>
</label><label ref="13"><input type="checkbox" id="ch4" ref="14" value="True">
</input><t class="TEXT\_CLASS" ref="None">DJfG</t></label><label ref="15"><input
type="checkbox" id="ch5" ref="16" value="True"></input><t class="TEXT\_CLASS"
ref="None">lI3Z</t></label></div><button id="subbtn" class="secondary-action"
ref="17">Submit</button></div></div></body>
```
{action: click, ref: 17}

Table 7: Success rate comparison of various models in MiniWoB tasks. Baseline results are borrowed from (Humphreys et al., 2022). Note that these are normalized between 0 and 1.

| TASK | Human | WebN-T5-3B | WebN-T5-3B (no history) | CC-Net (SL & RL) | CC-Net (SL) | World of bits (SL & RL) | Workflow guided exploration (SL & RL) | Learning to navigate the web (RL) | DOM-Q-Net (RL) | Workflow guided exploration (Augmented) | Learning to navigate the web (Augmented) | Aggregated SOTA (SL & RL) | Aggregated SOTA (Augmented) |
|---|---|---|---|---|---|---|---|---|---|---|---|---|---|
| bisect-angle | 0.92 | n/a | n/a | 0.97 | 0.29 | 0.8 | n/a | n/a | n/a | n/a | n/a | 0.8 | 0.8 |
| book-flight | 0.87 | 0 | 0 | 0.87 | 0 | 0 | 0 | n/a | n/a | 0 | 1 | 0 | 1 |
| chase-circle | 0.82 | n/a | n/a | 0.93 | 0.8 | 1 | n/a | n/a | n/a | n/a | n/a | 1 | 1 |
| choose-date-easy | 0.99 | 0.03 | 0.05 | 0.99 | 0.42 | n/a | n/a | n/a | n/a | n/a | n/a | n/a | n/a |
| choose-date-medium | 0.98 | 0 | 0 | 0.99 | 0.26 | n/a | n/a | n/a | n/a | n/a | n/a | n/a | n/a |
| choose-date | 0.97 | 0 | 0 | 0.97 | 0.12 | 0 | 0 | n/a | 1 | 0 | n/a | 1 | 1 |
| choose-list | 0.98 | 0.26 | 0.14 | 0.99 | 0.19 | 0.25 | 0.16 | 0.26 | n/a | 0.16 | 0.26 | 0.26 | 0.26 |
| circle-center | 0.96 | n/a | n/a | 0.97 | 0.36 | 0.98 | n/a | n/a | n/a | n/a | n/a | 0.98 | 0.98 |
| click-button-sequence | 0.94 | 1 | 1 | 1 | 0.47 | 0.22 | 0.99 | n/a | 1 | 1 | n/a | 1 | 1 |
| click-button | 0.98 | 1 | 0.96 | 1 | 0.78 | 0.62 | 1 | 1 | 1 | 1 | 1 | 1 | 1 |
| click-checkboxes-large | 0.87 | 0.22 | 0 | 0.71 | 0 | n/a | 0.68 | n/a | n/a | 0.84 | n/a | 0.68 | 0.84 |
| click-checkboxes-soft | 0.73 | 0.54 | 0.43 | 0.95 | 0.04 | n/a | 0.51 | n/a | n/a | 0.94 | n/a | 0.51 | 0.94 |
| click-checkboxes-transfer | 0.98 | 0.63 | 0.34 | 0.99 | 0.36 | n/a | 0.64 | n/a | n/a | 0.64 | n/a | 0.64 | 0.64 |
| click-checkboxes | 0.97 | 0.96 | 0.84 | 0.98 | 0.32 | 0.48 | 0.98 | n/a | 1 | 1 | n/a | 1 | 1 |
| click-collapsible-2 | 0.97 | 0 | 0.01 | 0.98 | 0.17 | 0.11 | 0.65 | n/a | n/a | 0.99 | n/a | 0.65 | 0.99 |
| click-collapsible | 0.99 | 0 | 0.01 | 1 | 0.81 | 0.98 | 1 | 1 | n/a | 1 | 1 | 1 | 1 |
| click-color | 0.97 | 0.27 | 0.23 | 1 | 0.82 | 0.23 | 1 | n/a | n/a | 1 | n/a | 1 | 1 |
| click-dialog-2 | 0.99 | 0.24 | 0.35 | 1 | 0.88 | 0.53 | 1 | n/a | n/a | 1 | n/a | 1 | 1 |
| click-dialog | 1 | 1 | 1 | 1 | 0.95 | 1 | 1 | 1 | 1 | 1 | 1 | 1 | 1 |
| click-link | 0.99 | 1 | 0.96 | 0.99 | 0.59 | 0.31 | 1 | 1 | 1 | 1 | 1 | 1 | 1 |
| click-menu-2 | 0.98 | n/a | n/a | 0.83 | 0.52 | 0.16 | n/a | n/a | n/a | n/a | n/a | 0.16 | 0.16 |
| click-menu | 0.97 | 0.37 | 0.38 | 0.94 | 0.22 | 0.13 | n/a | n/a | n/a | n/a | n/a | 0.13 | 0.13 |
| click-option | 0.99 | 0.87 | 0.78 | 0.99 | 0.21 | 0.28 | 1 | n/a | 1 | 1 | n/a | 1 | 1 |
| click-pie | 0.98 | 0.51 | 0.14 | 0.97 | 0.15 | 0.15 | 0.32 | 1 | n/a | 0.32 | 1 | 1 | 1 |
| click-scroll-list | 0.91 | 0 | 0 | 0.6 | 0.01 | 0.07 | n/a | n/a | n/a | n/a | n/a | 0.07 | 0.07 |
| click-shades | 0.91 | 0 | 0 | 1 | 0.04 | 0.27 | 0.22 | n/a | n/a | 0.99 | n/a | 0.27 | 0.99 |
| click-shape | 0.88 | 0.53 | 0.54 | 0.95 | 0.11 | 0.11 | 0.64 | n/a | n/a | 0.64 | n/a | 0.64 | 0.64 |
| click-tab-2-easy | 0.99 | n/a | n/a | 0.99 | 0.61 | n/a | n/a | n/a | n/a | n/a | n/a | n/a | n/a |
| click-tab-2-hard | 0.96 | 0.12 | 0.13 | 0.98 | 0.19 | n/a | n/a | n/a | n/a | n/a | n/a | n/a | n/a |
| click-tab-2-medium | 0.97 | n/a | n/a | 0.99 | 0.54 | n/a | n/a | n/a | n/a | n/a | n/a | n/a | n/a |
| click-tab-2 | 0.97 | 0.18 | 0.09 | 0.98 | 0.27 | 0.08 | 0.64 | n/a | 1 | 0.98 | n/a | 1 | 1 |
| click-tab | 0.99 | 0.74 | 1 | 1 | 0.95 | 0.97 | 0.55 | 1 | 1 | 1 | 1 | 1 | 1 |
| click-test-2 | 0.99 | 1 | 1 | 1 | 0.95 | 0.83 | 1 | n/a | 1 | 1 | n/a | 1 | 1 |
| click-test-transfer | 0.99 | n/a | n/a | 1 | 0.94 | n/a | n/a | n/a | n/a | n/a | n/a | n/a | n/a |
| click-test | 1 | 1 | 1 | 1 | 1 | 1 | 1 | n/a | 1 | 1 | n/a | 1 | 1 |
| click-widget | 0.83 | 1 | 0.97 | 1 | 0.56 | 0.34 | 0.93 | n/a | 1 | 0.93 | n/a | 1 | 1 |
| copy-paste-2 | 0.94 | n/a | n/a | 0.63 | 0.01 | 0 | n/a | n/a | n/a | n/a | n/a | 0 | 0 |
| copy-paste | 0.94 | n/a | n/a | 0.79 | 0.04 | 0 | n/a | n/a | n/a | n/a | n/a | 0 | 0 |
| count-shape | 0.82 | 0.41 | 0.43 | 0.85 | 0.21 | 0.18 | 0.59 | n/a | n/a | 0.76 | n/a | 0.59 | 0.76 |
| count-sides | 0.98 | n/a | n/a | 1 | 0.74 | 0.3 | n/a | n/a | n/a | n/a | n/a | 0.3 | 0.3 |
| drag-box | 0.99 | n/a | n/a | 1 | 0.61 | 0.31 | n/a | n/a | n/a | n/a | n/a | 0.31 | 0.31 |
| drag-cube | 0.99 | n/a | n/a | 0.79 | 0.23 | 0.18 | n/a | n/a | n/a | n/a | n/a | 0.18 | 0.18 |
| drag-item | 0.98 | n/a | n/a | 1 | 0.61 | n/a | n/a | n/a | n/a | n/a | n/a | n/a | n/a |
| drag-items-grid | 0.87 | n/a | n/a | 0.98 | 0.05 | 0.01 | n/a | n/a | n/a | n/a | n/a | 0.01 | 0.01 |
| drag-items | 0.93 | n/a | n/a | 0.99 | 0.13 | 0.41 | n/a | n/a | n/a | n/a | n/a | 0.41 | 0.41 |
| drag-shapes | 0.96 | n/a | n/a | 0.99 | 0.26 | 0.92 | n/a | n/a | n/a | n/a | n/a | 0.92 | 0.92 |
| drag-sort-numbers | 0.92 | n/a | n/a | 0.97 | 0.11 | 0.66 | n/a | n/a | n/a | n/a | n/a | 0.66 | 0.66 |
| email-inbox-delete | 0.99 | n/a | n/a | 1 | 0.22 | n/a | n/a | n/a | 1 | n/a | n/a | 1 | 1 |
| email-inbox-forward-nl-turk | 0.88 | 0.33 | 0.09 | 1 | 0 | n/a | n/a | n/a | n/a | n/a | n/a | n/a | n/a |
| email-inbox-forward-nl | 0.91 | 0.60 | 0.09 | 1 | 0 | n/a | n/a | n/a | n/a | n/a | n/a | n/a | n/a |
| email-inbox-forward | 0.96 | n/a | n/a | 1 | 0.01 | n/a | n/a | n/a | n/a | n/a | n/a | n/a | n/a |
| email-inbox-important | 0.99 | n/a | n/a | 1 | 0.3 | n/a | n/a | n/a | n/a | n/a | n/a | n/a | n/a |
| email-inbox-nl-turk | 0.93 | 0.23 | 0.26 | 1 | 0.05 | n/a | 0.77 | n/a | n/a | 0.93 | n/a | 0.77 | 0.93 |
| email-inbox-noscroll | 0.96 | n/a | n/a | 1 | 0.13 | n/a | n/a | n/a | n/a | n/a | n/a | n/a | n/a |
| email-inbox-reply | 0.91 | n/a | n/a | 1 | 0 | n/a | n/a | n/a | n/a | n/a | n/a | n/a | n/a |
| email-inbox-star-reply | 0.95 | n/a | n/a | 1 | 0.11 | n/a | n/a | n/a | n/a | n/a | n/a | n/a | n/a |
| email-inbox | 0.96 | 0.38 | 0.21 | 1 | 0.09 | 0.03 | 0.43 | n/a | 0.54 | 0.99 | n/a | 0.54 | 0.99 |
| enter-date | 0.97 | 0 | 0 | 1 | 0.02 | 0.61 | 0 | 1 | n/a | 0.96 | 1 | 1 | 1 |
| enter-password | 0.96 | 0.97 | 0.92 | 1 | 0.02 | 0 | 0.99 | 1 | 1 | 1 | 1 | 1 | 1 |
| enter-text-2 | 0.91 | n/a | n/a | 0.98 | 0.04 | 0 | n/a | n/a | n/a | n/a | n/a | 0 | 0 |
| enter-text-dynamic | 0.97 | 0.98 | 0.92 | 1 | 0.39 | 1 | 1 | 1 | 1 | 1 | 1 | 1 | 1 |
| enter-text | 0.98 | 0.89 | 0.99 | 1 | 0.35 | 0 | 1 | n/a | 1 | 1 | n/a | 1 | 1 |
| enter-time | 0.98 | 0 | 0.01 | 0.97 | 0.04 | 0.08 | 0.52 | n/a | n/a | 0.9 | n/a | 0.52 | 0.9 |
| find-midpoint | 0.94 | n/a | n/a | 0.97 | 0.35 | 0.31 | n/a | n/a | n/a | n/a | n/a | 0.31 | 0.31 |
| find-word | 0.96 | n/a | n/a | 0.88 | 0.05 | 0 | n/a | n/a | n/a | n/a | n/a | 0 | 0 |
| focus-text-2 | 0.99 | 1 | 1 | 1 | 0.96 | 0.83 | 1 | n/a | 1 | 1 | n/a | 1 | 1 |
| focus-text | 1 | 1 | 1 | 1 | 0.99 | 0.95 | 1 | n/a | 1 | 1 | n/a | 1 | 1 |
| grid-coordinate | 0.87 | 0.49 | 0.42 | 1 | 0.66 | 0.26 | 1 | n/a | 1 | 1 | n/a | 1 | 1 |
| guess-number | 0.99 | 0 | 0 | 1 | 0.21 | 0.2 | 0 | n/a | n/a | 0 | n/a | 0.2 | 0.2 |
| highlight-text-2 | 0.97 | n/a | n/a | 1 | 0.4 | 0.13 | n/a | n/a | n/a | n/a | n/a | 0.13 | 0.13 |
| highlight-text | 0.97 | n/a | n/a | 1 | 0.51 | 0.9 | n/a | n/a | n/a | n/a | n/a | 0.9 | 0.9 |
| identify-shape | 0.98 | 0.88 | 0.89 | 1 | 0.68 | 0.36 | 0.9 | n/a | n/a | 1 | n/a | 0.9 | 1 |
| login-user-popup | 0.94 | 0.72 | 0.40 | 1 | 0.02 | n/a | n/a | n/a | n/a | n/a | n/a | n/a | n/a |
| login-user | 0.96 | 0.82 | 0.64 | 1 | 0 | 0 | 0.99 | 1 | 1 | 1 | 1 | 1 | 1 |
| moving-items | 0.18 | n/a | n/a | 0.88 | 0.13 | 0.78 | n/a | n/a | n/a | n/a | n/a | 0.78 | 0.78 |
| multi-layouts | 0.95 | 0.83 | 0.48 | 1 | 0 | n/a | 0.99 | n/a | n/a | 1 | n/a | 0.99 | 1 |
| multi-orderings | 0.96 | 0.88 | 0.64 | 1 | 0 | n/a | 0.05 | n/a | n/a | 1 | n/a | 0.05 | 1 |
| navigate-tree | 0.98 | 0.91 | 0.99 | 0.99 | 0.32 | 0.2 | 0.99 | 1 | 1 | 0.99 | 1 | 1 | 1 |
| number-checkboxes | 0.96 | n/a | n/a | 0.99 | 0 | 0.16 | n/a | n/a | n/a | n/a | n/a | 0.16 | 0.16 |
| read-table-2 | 0.95 | n/a | n/a | 0.94 | 0 | 0 | n/a | n/a | n/a | n/a | n/a | 0 | 0 |
| read-table | 0.97 | n/a | n/a | 0.97 | 0.01 | 0 | n/a | n/a | n/a | n/a | n/a | 0 | 0 |
| resize-textarea | 0.94 | n/a | n/a | 1 | 0.27 | 0.11 | n/a | n/a | n/a | n/a | n/a | 0.11 | 0.11 |
| right-angle | 0.87 | n/a | n/a | 0.98 | 0.26 | 0.38 | n/a | n/a | n/a | n/a | n/a | 0.38 | 0.38 |
| scroll-text-2 | 0.97 | n/a | n/a | 1 | 0.88 | 0.96 | n/a | n/a | n/a | n/a | n/a | 0.96 | 0.96 |
| scroll-text | 0.97 | n/a | n/a | 0.96 | 0.04 | 0 | n/a | n/a | n/a | n/a | n/a | 0 | 0 |
| search-engine | 0.97 | 0.34 | 0.34 | 1 | 0.15 | 0 | 0.26 | n/a | 1 | 0.99 | n/a | 1 | 1 |
| simon-says | 0.62 | n/a | n/a | 0 | 0.02 | 0.28 | n/a | n/a | n/a | n/a | n/a | 0.28 | 0.28 |
| simple-algebra | 0.86 | n/a | n/a | 0.75 | 0.03 | 0.04 | n/a | n/a | n/a | n/a | n/a | 0.04 | 0.04 |
| simple-arithmetic | 0.96 | n/a | n/a | 0.86 | 0.38 | 0.07 | n/a | n/a | n/a | n/a | n/a | 0.07 | 0.07 |
| social-media-all | 0.89 | 0 | 0 | 0.75 | 0 | n/a | 0.01 | n/a | n/a | 0.01 | 1 | 0.01 | 1 |
| social-media-some | 0.91 | 0.02 | 0 | 0.85 | 0.01 | n/a | 0.01 | n/a | n/a | 0.42 | n/a | 0.01 | 0.42 |
| social-media | 0.96 | 0.21 | 0.24 | 0.9 | 0.03 | 0.23 | 0.39 | n/a | 1 | 1 | n/a | 1 | 1 |
| terminal | 0.88 | n/a | n/a | -0.01 | 0 | 0 | n/a | n/a | n/a | n/a | n/a | 0 | 0 |
| text-editor | 0.88 | n/a | n/a | 0.98 | 0.11 | 0.01 | n/a | n/a | n/a | n/a | n/a | 0.01 | 0.01 |
| text-transform | 0.86 | n/a | n/a | 0.6 | 0.19 | 0 | n/a | n/a | n/a | n/a | n/a | 0 | 0 |
| tic-tac-toe | 0.71 | 0.48 | 0.40 | 0.83 | 0.32 | 0.34 | 0.37 | n/a | n/a | 0.47 | n/a | 0.37 | 0.47 |
| unicode-test | 0.99 | n/a | n/a | 1 | 0.86 | n/a | n/a | n/a | n/a | n/a | n/a | n/a | n/a |
| use-autocomplete | 0.98 | 0.22 | 0.15 | 1 | 0.07 | 0 | 0.78 | n/a | n/a | 0.98 | n/a | 0.78 | 0.98 |
| use-colorwheel-2 | 0.94 | n/a | n/a | 0.95 | 0.38 | 1 | n/a | n/a | n/a | n/a | n/a | 1 | 1 |
| use-colorwheel | 0.9 | n/a | n/a | 0.98 | 0.68 | 1 | n/a | n/a | n/a | n/a | n/a | 1 | 1 |
| use-slider-2 | 0.97 | n/a | n/a | 0.95 | 0.03 | 0.15 | n/a | n/a | n/a | n/a | n/a | 0.15 | 0.15 |
| use-slider | 0.98 | n/a | n/a | 0.91 | 0.18 | 0.51 | n/a | n/a | n/a | n/a | n/a | 0.51 | 0.51 |
| use-spinner | 0.98 | 0.07 | 0.05 | 1 | 0.47 | 0.17 | 0.04 | n/a | n/a | 0.04 | n/a | 0.17 | 0.17 |
| visual-addition | 0.97 | n/a | n/a | 0.99 | 0.36 | 0.01 | n/a | n/a | n/a | n/a | n/a | 0.01 | 0.01 |

Table 8: Resource requirements and running time of LLMs.

| Model Name | Model Size | TPU version | Batch size | Input sequence length | Examples per sec (training) | Examples per sec (inference) |
|---|---|---|---|---|---|---|
| PaLM | 62B | TPU v4 | 8 | 1920 | 9.313 | 30.51 |
| PaLM | 8B | TPU v4 | 32 | 1920 | 64.4 | 184.3 |
| T5 | 3B | TPU v4 | 128 | 512 | 163.8 | 734.5 |
| LaMDA | 1B | TPU v2 | 128 | 512 | 363.1 | 1416 |

