# OpenReview forum: "UNDERSTANDING HTML WITH LARGE LANGUAGE MODELS"
_ICLR.cc/2023/Conference — Submitted to ICLR 2023_

### Official Review · Reviewer_d1QN · 2022-10-24

**Confidence:** 3
**Correctness:** 3
**Technical Novelty And Significance:** 2
**Empirical Novelty And Significance:** 3
**Recommendation:** 5

**Clarity, Quality, Novelty And Reproducibility:**

- Clarity: The paper is generally clear and easy to understand, some tables and figures in experiments are confusing or hard to read.

- Novelty: The new task is somewhat interesting and novel. The modeling part is not that novel.

- Reproducibility: Authors claim to open-source the Description Generation dataset.

**Strength And Weaknesses:**

Pros: The direction of HTML understanding is increasingly important, as recent work has started to show the promise of utilizing the semantics of HTML and web navigation. The paper adds solid contributions to this direction by creating a novel and interesting task, and conducting many empirical studies with insights about architecture, (sub-linear) scaling effect, importance of context window/action history, etc.

Cons:

- On the proposed Description Generation task, authors only spend two paragraphs presenting results, where WebD-T55-3B can obtain 90.8% accuracy (exact string match??). Does it mean the task is too easy? How to justify it from the existing semantic classification task, as in Figure 1(a) they look similar? Why in Table 3, dev numbers are much lower than test set? I feel the new task/dataset needs more explanation and justification.

- The modeling part lacks novelty, just basic pre-training + fine-tuning on LLMs, and seems there's no model design based on the features of HTML?

- On MiniWoB, it's not very surprising that using a larger model with pre-training, you can beat smaller models from scratch. Instead, the fact that LLMs are worse than RL performances should be discussed more -- does it mean the task is bad, or RL is important (but it's hard to do RL for LLM)?

- (Minor) In intro authors claim "no processing of HTML", but there is still some pre-processing?


**Summary Of The Paper:**

The paper proposes two main contributions to the direction of HTML modeling/understanding:

1. A new task and dataset, **Description Generation**, derived from CommonCrawl with 85K (HTML, element, description) tuples.

2. Experiments across a suite of three HTML tasks (Description Generation, semantic classification from Gur et al., MiniWob web navigation), investigating the importance of pre-training corpora (natural language and/or HTML), model architecture (encoder-only, decoder-only, encoder-decoder), the scaling effect, and so on. The main finding is the effectiveness of large language models (LLM) pre-trained on natural language and fine-tuned on HTML task data, with T5-like encoder-decoder architecture being the best.

**Summary Of The Review:**

I believe the paper makes solid empirical contributions by introducing a new task, and studying many LLM variants on three HTML tasks, with some insights about what's important for modeling HTML. My main concern is that the new task is not clearly justified, and the modeling part might be too standard -- there is no HTML-specific model design involved, just studying basic LLMs. I think the paper can benefit from more evidence about the value of the new benchmark (e.g. what new insights/modeling capabilities it enables), or more interesting modeling.

---

> ### Author Response · Authors · 2022-11-18
> **Thank you for the detailed feedback and suggestions.**
>
> **Description Generation: Definition and Difficulty?**
>
> We agree that the observed performance for description generation suggests that baseline performance is high. Nevertheless, it is an important task - the ability to accurately associate  label texts with inputs is an important prerequisite for any form-filling agent, and has potential applications for improving accessibility as well. Furthermore, for autonomous agents the bar for strong performance for such a task is often high - 85% would not be considered sufficiently strong performance for automation tasks involving financial transactions.
>
> On the question regarding the accuracy metric for description generation - examples  are considered “correct” if they match exactly. We supplement this stringent definition by reporting BLUE and ROUGE-1 scores which reflect partial matches between ground truth description and generated text.
>
> **Description Generation vs Semantic Classification**
>
> The reviewer is correct to point out that description generation is related to (but not the same as) semantic classification. In many use cases, the ultimate pre-requisite for form-filling agents is accurate semantic classification. On the other hand, semantic class definitions will vary between use cases.
>
> Description generation is a more task-agnostic capability. For some use cases one could conceivably group related descriptions and assign them to a shared semantic class if desired. Defining the description task more granularly allows two things: to have a general self-supervision procedure (using the “for” tag for the minority of cases where it’s available) and to provide a publicly available dataset that can be used to train other models without being tied to the semantic categorization of a specific, narrow use case.
>
> We thank the reviewer for pointing out the potential dev/test split issue in Table 3. We reviewed common crawl’s grouping of pages domain and identified an issue with oversampling domains leading to a skewed dev/test split. We created new dev and test splits by making sure that statistics are closer while domains are still disjoint. We also updated the Table-3 with updated results.
>
> **Novelty**
>
> We thank the reviewer for the feedback regarding significance and novelty. We agree our approach leverages existing LLM architectures, and our focus is not to propose a new architecture tailored towards HTML. We also agree that pre-training/fine-tuning is mature in supervised learning tasks for NLP and are not a novel contribution of this work.
>
> The focus of our technical contributions is to advance the integration of LLMs with autonomous web agents. It’s only in the last year that researchers have begun to utilize LLMs outside of NLP and integrate them as core capabilities in autonomy. In this context LLMs are reasoning engines for sequential decision making agents interacting with environments.
>
> The present work is the first in the research literature to embed an LLM and train it as an agent in the context of autonomous web navigation. This requires new implementations to adapt LLM training for behavior cloning in addition to designing interfaces for integrating text generation into a perception-compute-action cycle operating in a stateful web environment. Although text understanding remains relevant to perception, modeling considerations for web tasks are different from standard NLP due to the sequential aspect of autonomous navigation tasks.
>
> Our implementation allows us to answer novel questions of trade-offs among various model characteristics that are relevant to the integration of LLMs in autonomy. We demonstrate how the learning efficiency of LLMs translated to agent-based tasks in web navigation, provide public releases of models and datasets alongside a new self-supervised description generation method, and show how smaller bidirectional models can outperform larger autoregressive models. We believe these contributions expand the scope of language models and connects their unique capabilities with autonomous agents for the web.

---

> > ### Author Response · Authors · 2022-11-18
> > **Further details.**
> >
> > **LLMs and RL**
> >
> > The comparisons in figure 2 are not intended to propose LLMs as an alternative to RL. In fact, our current hypothesis is that LLMs trained with RL will improve performance further. The behavior cloning implementation establishes a future baseline for embedding an LLM in an autonomous agent for this task. Extending LLM-based agents to incorporate RL is a natural follow-up, however, it requires additional development to scale simulator environment training to accommodate LLMs.
> >
> > On the specifics of comparisons in figure 2 - in the case of comparison to CC-Net (SL + RL) the 2.4 million example dataset (not publicly available) is at least one contributing factor. In the case of WGE, the model has strong inductive biases for sampling generated actions by using a grammar and workflow generation for control.
> >
> > One direction for future work is to incorporate stronger inductive biases for generated actions than the flexible dictionary generation mechanism in this work. For example, actions from a fixed grammar/workflow could be scored individually instead of parsing generated text to actuate browser actions. Alternatively, generated text itself could have more structured outputs such as code generation.
> >
> > We expect combining these strategies of stronger inductive biases, alternative action generation mechanisms, and RL provide opportunities for further improving LLM-based autonomous web agents in the future.
> >
> > **Processing of HTML**
> >
> > We amend the passage in related work to read “HTML in text form with minimal processing”.

---

### Official Review · Reviewer_wKTj · 2022-10-24

**Confidence:** 4
**Correctness:** 3
**Technical Novelty And Significance:** 3
**Empirical Novelty And Significance:** 3
**Recommendation:** 5

**Clarity, Quality, Novelty And Reproducibility:**

Clarity: Most things are explained clearly either in the main paper or the appendix, except for the few points that are raised in the Weaknesses above.
Quality: well-written work.
Novelty: The novelty is primarily empirical, and the technical approach is essentially an application of known techniques (the standard pretrain-finetune paradigm of LLMs).
Reproducibility: The Supplementary Material only provides the data file. It is better to make the codes available for reproducing these results.

**Strength And Weaknesses:**

Strengths:
1. The paper is well-written and easy to follow.
2. The authors conduct comprehensive evaluations and analyses over a range of architectures, dataset sizes, and baselines to make the experimental conclusion convincing.
3. The authors create and open-source a new dataset/benchmark for HTML understanding.

Weaknesses:
1. The novelty is primarily empirical, and the technical approach is essentially an application of known techniques (the standard pretrain-finetune paradigm of LLMs).
2. I examined the supplementary material provided by the authors and only found the data files. It would be better if there were codes to prove its reproducibility.

**Summary Of The Paper:**

The paper investigates whether LLMs can be applied to HTML understanding to produce better-performing, more sample-efficient HTML understanding models without the need for custom NN architecture design. To achieve this goal, the authors present a suite of three benchmarking tasks for HTML understanding: (1) Semantic Classification of HTML elements, (2) Description Generation for HTML inputs, and (3) Autonomous Web Navigation of HTML pages. The authors find that a) pretraining is critical for the performance and can reduce labeled data requirements / improve sample efficiency up to 200x; b) model architecture is the second most important factor, and T5 models with bi-directional attention and encoder-decoder architecture performs the best across the board; c) given a choice, model size should be evaluated in the context of the models training and inference performance, as the model size sub-linearly predicts its performance.

**Summary Of The Review:**

In my opinion, the audience could gain insights from the thorough empirical analysis of HTML understanding from this paper. However, the lack of technical novelty & contribution makes this work slightly below the acceptance bar of the conference. Overall, I recommend a weak reject.

---

> ### Author Response · Authors · 2022-11-18
> **Thank you for the useful feedback.**
>
> We appreciate the feedback. Regarding the implementation - we open sourced our checkpoints for WebN-T5-3B and WebN-T5-large models (https://drive.google.com/drive/folders/1aNXHyj-PU3hJcaofWqabRh3urmr4H_g-). The evaluation code will be open sourced in the coming weeks.
>
> We agree that for supervised learning NLP tasks, fine tuning an LLM is standard - though we emphasize here that the current work is not only finetuning as done in NLP.
>
> The broader context for this work is that it has only been in the last year that the field has begun to understand the role of LLMs outside of NLP and in the context of autonomy. LLMs can serve as reasoning engines for autonomous agents that interact with external stateful environments (for embodied agents, SayCan is one example of this emerging viewpoint).
>
> This paper is the first to embed multiple large (several hundred million up to 62B-parameter) language models and train it as an agent in the context of autonomous web navigation. This requires a technical implementation of both adapting the fine tuning paradigm for behavior cloning of an agent as well as embedding and defining how LLMs fit into a perception-compute-action cycle of a web navigation environment spanning a broad range of tasks. These tasks subsume NLP as a perception requirement, but are very different in nature due to the sequential decision-making aspect of autonomous navigation.
>
> Our implementation allows us to answer novel questions of trade-offs among various model characteristics that are relevant to the integration of LLMs in autonomy. We demonstrate how the learning efficiency of LLMs translated to agent-based tasks in web navigation, provide public releases of models and datasets alongside a new self-supervised description generation method, and show how smaller bidirectional models can outperform larger autoregressive models. We believe these contributions expand the scope of language models and connect their unique capabilities with autonomous agents for the web.

---

### Official Review · Reviewer_hMUi · 2022-10-25

**Confidence:** 3
**Correctness:** 2
**Technical Novelty And Significance:** 2
**Empirical Novelty And Significance:** 3
**Recommendation:** 5

**Clarity, Quality, Novelty And Reproducibility:**

- Clarity: The paper is easy to follow as the content of the paper is simple.
- Quality: The experiments seem to be well-done.
- Novelty: The novelty is limited in terms of methodology. It is mostly an empirical study for large LLMs on HTML texts.
- Reproducibility: Some details are missing. Either code or more detailed experimental setup should be provided in the supplementary material

**Strength And Weaknesses:**

## Strength
- The paper is easy to follow.
- The study of how LLMs perform on raw HTML texts is interesting and novel.

## Weaknesses
- Models that are trained on task specific data to compare against is questionable. I expect to see more task-specific models such as StructuralLM [1], MarkupLM [2] that respect the tree structure of HTML (or some other methods mentioned in the paper) to be evaluated against. However, the results seem to focus on LLMs only. I don't see the point of compare different variants of LLMs with or without pre-training. Basically, I need to see the following question answered: "For specific web tasks, do I want to use a HTML specific model trained on the small dataset or just to fine-tune LLMs"?

[1] Li, C., Bi, B., Yan, M., Wang, W., Huang, S., Huang, F. and Si, L., 2021. StructuralLM: Structural Pre-training for Form Understanding. arXiv preprint arXiv:2105.11210.
[1] Li, J., Xu, Y., Cui, L. and Wei, F., 2021. MarkupLM: Pre-training of text and markup language for visually-rich document understanding. arXiv preprint arXiv:2110.08518.

**Summary Of The Paper:**

The paper studies how pre-trained large language models (LLMs) perform on three HTML understanding tasks: (1) semantic classification of HTML elements, (2) description generation for HTML inputs and (3) autonomous web navigation of HTML pages. It found that pre-trained LLMs can work on these tasks effectively after fine-tuning. They require less task-specific data while perform better than the previous best supervised model.

**Summary Of The Review:**

I like the idea of studying LLMs on HTML texts. However, it seems to me that the experiment setup (re. models to compare against) cannot support the main claim of why one would like to do so. Besides, the technical novelty of the paper is limited for a ML conference like ICLR; a NLP conference might be a better fit for this submission.

---

> ### Author Response · Authors · 2022-11-18
> **Thank you for the valuable feedback and related work.**
>
> We appreciate the reviewer’s feedback regarding novelty and interest in the use of LLMs for  HTML text as well as highlighting relevant literature on task-specific models. We have updated the related work section accordingly.
>
> The reviewer raises the question of comparison to these task-specific models. Below, we summarized and compared them to Web-LLMs that we studied. We also conducted new experiments using MarkupLM to better understand their applicability.
>
> **Novelty**
>
> In regards to technical contribution and novelty it is important to consider this work not just in the context of NLP-oriented processing of HTML data, but as a step towards embedding LLMs in autonomous agents for the web.
>
> Although the field of natural language processing (NLP) has mature techniques for LLM pre-training and finetuning for supervised learning, it has only been in the last year that researchers have begun to incorporate LLMs more broadly in the context of autonomy. For autonomous agents, LLMs serve as reasoning engines for real time sequential decision-making in stateful environments.
>
> The present work is the first in the research literature to embed an LLM and train it as an agent in the context of autonomous web navigation. This requires a technical implementation of both adapting the fine tuning paradigm for behavior cloning of an agent as well as designing how LLMs fit into a perception-compute-action cycle of a web navigation environment spanning a broad range of tasks. Although understanding text remains relevant from a perception standpoint, the modeling considerations are very different from standard NLP tasks due to the sequential decision making aspect of autonomous navigation.
>
> Our implementation which integrates an LLM as a web agent allows us to answer novel questions of trade-offs among various model characteristics that are relevant to the integration of LLMs in autonomy. We demonstrate how the learning efficiency of LLMs translated to agent-based tasks in web navigation and show how smaller bidirectional models can outperform larger autoregressive models. We have updated the introduction to clarify these methodological contributions.
>
> **StructuralLM**
>
> StructuralLM is an approach specifically tailored for document understanding (i.e., combinations of images and text), and thus makes several simplifying assumptions for its model that limit its applicability to HTML understanding (i.e., trees of elements with a richer structure and functionality):
>
>
> StructuralLM is trained only on the textual content of a document - the markup information is ignored. For example, any input field or dropdown in a document would be missing from the model inputs. All of the tasks we study require knowledge of this information. For example, in autonomous navigation the model needs to interact with input elements (e.g. text, checkboxes, dropdowns) such as username and password in the login-user task in MiniWoB. Typically, a “type” action with a reference to an element and a text argument is generated by the model. Without knowing which input elements are available in the page, it is impossible to generate a reference to any input element.
>
> **MarkupLM**
>
> While MarkupLM is better tailored for understanding HTML pages, it has similar drawbacks as StructuralLM in that it focuses solely on text while ignoring the markup. It is evaluated on NLP-like tasks such as QA or entity classification where understanding page content is paramount, whereas we focus on HTML understanding tasks such as autonomous navigation where both content and the page’s layout structure need to be understood.
>
> We perform a quantitative evaluation of MarkupLM on our tasks to understand how significant these limitations are. We fine-tune the MarkupLM-base model on the semantic classification task, using the same setup as other WebC models but with the suggested hyperparameters that authors provided in the MarkupLM work. On development and test sets, MarkupLM-base achieves 65% and 66% accuracy, respectively. These results are more than 16% lower compared to similar size WebC-BERT-base results that we report in our work.
>
> Our analysis suggests that although domain specific models may be suitable for processing HTML for NLP tasks, the generality, flexibility, and sample efficiency LLMs provide advantages when considering LLMs for autonomous agents for the web.

---

> > ### Comment · Reviewer_hMUi · 2022-11-19
> > **Thanks for the response**
> >
> > > ... autonomous agents for the web.
> >
> > This is a very good point.
> >
> > > While MarkupLM is better tailored for understanding HTML pages, it has similar drawbacks as StructuralLM in that it focuses solely on text while ignoring the markup.
> >
> > Can you clarify this criticism to MarkupLM?
> > As far as I know, MarkupLM does not ignore markup; see how tags are used in figure 2 of their paper.

---

> > > ### Author Response · Authors · 2022-11-19
> > > **Thank you for the prompt response!**
> > >
> > > We would like to clarify our sentence that MarkupLM focuses solely on text and structure of text while ignoring everything else in the markup. In Figure-2 of their paper, text extractor finds natural language words in the HTML and xpath extractor generates xpaths of only these words.
> > >
> > > We provide an additional example in the Appendix A.7 that illustrates our point better. We used the MarkupLM to process the sample HTML snippet in Figure-1(b) of our paper. The MarkupLM ignores all input elements, both username and password, and generates *<s>Email Address Enter Password: Please enter your password.</s>* which is the text input to the MarkupLM Transformer. Classifying this text as username or password is not possible without the additional context on which input element is the salient element (in this context it is the username). We also provide the code to reproduce our result.
> > >
> > > Please let us know if there are any remaining concerns.

---

> > > > ### Comment · Reviewer_hMUi · 2022-12-02
> > > > **Clarification question**
> > > >
> > > > > ... solely on text and structure of text while ignoring everything else in the markup ...
> > > >
> > > > The XPath contains tag types (e.g., title v.s. span) so this cannot be true.
> > > > But I guess what you really mean is that the tag type is the only thing used within the tag but no other information within the tag (such as type or id) is used. Is my understand correct?

---

> > > > > ### Author Response · Authors · 2022-12-02
> > > > > **Thanks for the response!**
> > > > >
> > > > > As you pointed out, one limitation of MarkupLM is that XPaths contain only "tag" and other attributes such as id, type, value, class, etc. are ignored. This is important as these attributes can have crucial information like "id" can tell if an "input" is "username" or "password".
> > > > >
> > > > > Another limitation that we wanted to highlight in our previous response is that "only XPaths of textual elements are used while XPaths of all other elements are ignored". If an element, such as an input or button, is not a natural language text, its XPath is not included as input to the model. For example, take the HTML snippet  in Figure-1(b):
> > > > >
> > > > > `<div><label class="form-label" for="uName">Email Address
> > > > > </label><label class="form-label" for="pass">Enter Password:
> > > > > </label></div><div><input type="email" id="uName" target><input
> > > > > type="password" id="pass"><span class="hidden">Please enter your password.
> > > > > </span></div>`
> > > > >
> > > > > After preprocessing, the input to the model is (padding and other special tokens are omitted for bravity):
> > > > >
> > > > > **Text:**     *Email Address Enter Password : Please enter your password .*
> > > > >
> > > > > **XPath:**  *div/label div/label div/label div/label div/label div/span div/span div/span div/span div/span*
> > > > >
> > > > > Note that all non-textual elements, including "inputs", along with their XPaths are ignored. In Appendix A.7, we also shared a code snippet to reproduce our analysis.
> > > > >
> > > > > Please let us know if you have any other questions.

---

> > > > > > ### Comment · Reviewer_hMUi · 2022-12-02
> > > > > > **Thanks for the clarification**
> > > > > >
> > > > > > I now understand your point and do see the merit of the proposed model (motivation and technical difference).
> > > > > > I raised my score accordingly.

---

### Official Review · Reviewer_5MM1 · 2022-10-30

**Confidence:** 4
**Correctness:** 3
**Technical Novelty And Significance:** 2
**Empirical Novelty And Significance:** 2
**Recommendation:** 6

**Clarity, Quality, Novelty And Reproducibility:**

The submission is well written and easy to understand. The three canonical tasks are described well and the adaptation of the various LLM for building models for these tasks are well explained. The proposed solution is simple and appears to be effective for the tasks considered and the datasets chosen. There is not much novelty in methodological aspects and the work is primarily empirical in nature. Experiments are designed well and should be easy to reproduce. Datasets used in the experiments have been promised to be released. The work should be interesting for practitioners.





**Strength And Weaknesses:**

Strengths:
	1. That pre-trained natural language LLM can be effective for tasks involving HTML pages  is interesting and can potentially find use in several interesting practical applications.
	2. As no retraining of LLM with large HTML datasets is necessary, models for tasks  involving HTML pages can be developed quickly and less expensively.
	3. That raw HTML text can be used as input without needing parsing is an advantage.
        4. Experimental results are very encouraging and validate the claim that pretrained LLMs can be effective for the three tasks.

Weaknesses:
        1. It is claimed that these three tasks require understanding of both structure and content of the web-page. While it is easy to see that textual content plays a key role in each of the three tasks, the role played by the structure of the web-page is not clear. It can be argued that no significant HTML structure analysis or understanding is needed for these tasks. For example, in Semantic Classification, what is most important for classifying HTML element 'input' into, say, 'username' is the value of its two attributes,  'type' and 'id'. As these attributes are in the close neighbourhood of 'input', parsing of HTML is not strictly necessary. Therefore, it might a good idea to do some experiments that demonstrate unequivocally the need for HTML structure analysis or understanding in these tasks. One such experiment could be to map all HTML tags in the web-page except the salient tags to the same token (say, ***) so that the input is now a sequence of salient tags, and ***.
      2. There is not much novelty in the methodological aspects of the work.


**Summary Of The Paper:**

This work addresses the problem of using large language models for understanding HTML. Unlike prior work which attempt to solve this problem using dedicated architectures and training procedures and/or large HTML corpora, this work employs large language models pretrained on natural language text and evaluates their performance on three HTML understanding tasks - Semantic Classification of HTML elements, Description Generation for HTML inputs, and Autonomous Web Navigation of HTML pages, thus potentially eliminating the need for dedicated architectures and training procedures. Further, using only a small HTML corpus for finetuning a pretrained LM, the work reports encouraging results compared to LMs trained exclusively on the task dataset.


The key question asked by this work is can off-the-shelf LLM trained on a large text corpus be used in tasks that require some level of understanding of HTML. As canonical tasks in HTML understanding, the work looks at three tasks.

In Semantic Classification, the ask from the model is to classify a salient HTML element into one of a set of role categories that are commonly used in automated form-filling applications. E.g. address, email, password.

In Description Generation, the ask from the model is to, given a HTML snippet as the input, extract a small text sequence from the snippet as the natural language description of the snippet.

In Autonomous Web Navigation, the ask from the model is to, given a HTML page and a natural language command as the input, identify the appropriate HTML elements and the actions on those elements that would satisfy the command.


The work tests the idea of using pre-trained LLM for the three canonical tasks with several pretrained LLMs with different architecture  encoder-only, encoder-decoder, or decoder-only, different model size, and training data. Best results are obtained with encoder-decoder architectures with bi-directional attention.

The input to the models is the raw HTML text sequence. However, when the sequence is too big to fit into the context window of LLM, a snippet of appropriate size is extracted using a heuristic algorithm.

The work uses MiniWoB benchmark (demonstrations like email forwarding and social media interactions) for Autonomous Web Navigation task, a new dataset consisting of URLs from the real shopping websites for Semantic Classification, and a dataset derived from CommonCrawl for Description Generation.



**Summary Of The Review:**

This work asks the question can off-the-shelf LLM trained on natural language text be used effectively for tasks that involve HTML pages. It proposes three tasks as canonical tasks in understanding HTML. It employs a variety of LLM to build models for the three tasks using a small amount of HTML data for fine tuning. It shows that LLM does help these tasks significantly. One key question not answered in this context is how much of HTML structure analysis and understanding is truly required for these questions.

---

> ### Author Response · Authors · 2022-11-18
> **Thank you for the valuable comments.**
>
> We thank the reviewer for highlighting the applicability and benefits of LLMs for HTML and promising experimental findings. We also appreciate the perspective and constructive feedback. We discuss each of them below.
>
> **Is Structural Interpretation Essential?**
>
> We agree that a better understanding of the model’s dependence on page structure would be informative. Following these suggestions, we conduct an ablation study to examine the sensitivity of model performance to preserving structural information.
>
> To do so, we devise an ablation that evaluates the model’s performance on HTML input with critical structure components removed. We kept the order of elements and their attributes fixed while corrupting the nesting structure by removing closing tags. Removing closing tags corresponds to a valid traversal (BFS) and keeps the order of elements the same as the text based input.
>
>
> As a simple example, <div id=”form”><div><input id=”username”></div></div> would be converted into <div id=”form”><div><input id=”username”>.
>
> We evaluated the trained WebN-T5-3B model on the same set of synthetic websites from the MiniWoB benchmark with this aspect of structure removed from the HTML pages. WebN-T5-3B achieves a 45.4% success rate, 6% lower than before, suggesting that WebN-T5-3B is at least partially utilizing the DOM topology.
>
> We include this ablation in Appendix A.6.
>
> **Methodological and Technical Novelty**
>
> We thank the reviewer for the feedback regarding significance and novelty. We agree our approach leverages existing LLM architectures, and our focus is not to propose a new architecture tailored towards HTML. We also agree that pre-training/fine-tuning is mature in supervised learning tasks for NLP and are not a novel contribution of this work.
>
> The focus of our technical contributions is to advance the integration of LLMs with autonomous web agents. It’s only been in the last year that researchers have begun to utilize LLMs outside of NLP and integrate them as core capabilities in autonomy. In this context, LLMs are reasoning engines for sequential decision-making agents interacting with environments.
>
> The present work is the first in the research literature to embed an LLM and train it as an agent in the context of autonomous web navigation. This requires new implementations to adapt LLM training for behavior cloning in addition to designing interfaces for integrating text generation into a perception-compute-action cycle operating in a stateful web environment. Although text understanding remains relevant to perception, modeling considerations for web tasks are different from standard NLP due to the sequential aspect of autonomous navigation tasks.
>
> Our implementation allows us to answer novel questions of trade-offs among various model characteristics that are relevant to the integration of LLMs in autonomy. We demonstrate how the learning efficiency of LLMs translated to agent-based tasks in web navigation, provide public releases of models and datasets alongside a new self-supervised description generation method, and show how smaller bidirectional models can outperform larger autoregressive models.
>
> We believe these contributions expand the scope of language models and connect their unique capabilities with autonomous agents for the web.

---

### Author Response · Authors · 2022-11-18
**Summary of changes**

We thank the reviewers for their thoughtfully considered comments. We agree with reviewer hMUI that “how LLMs perform on raw HTML is both interesting and novel” and appreciate the positive feedback regarding clear and understandable presentation of comprehensive results.

Their constructive input has also helped us design clarifying follow-up experiments. Furthermore, we have refined the introduction to better contextualize this work in the broader landscape of LLMs for autonomous agents.

A summary of the changes made in response to their feedback are as follows:

1. An ablation study was added as Appendix A.6 in response to reviewer 1 (5MM1s) to assess the sensitivity of performance to preserving/eliminating HTML structure.


2. A comparison using a task-specific model (MarkupLM) was added as Appendix A.7 in response to reviewer 2’s (hMUi) suggestions.


3. We added a passage to the introduction to contextualize the core technical contribution of the work. We clarify that although the practice of pre-training/fine-tuning LLMs is mature for supervised learning in NLP, the integration of LLMs for sequential decision making in autonomous agents is a nascent subject with foundational open questions. To our knowledge, this work is the first research publication which trains/integrates multiple LLMs into the perception-compute-action cycle of agents interacting with a stateful web environment.


4. Minor clarifications - refer to processing of HTML as “minimal” rather than “none” due to the use of snippets in response to reviewer 4 (“2. Related Work”) and the relevancy of structure to apply to all 3 tasks rather than semantic classification alone in response to reviewer 1 (“4. Canonical Understanding…”).

5. We open sourced our checkpoints for the WebN-T5-3B and WebN-T5-large models (https://drive.google.com/drive/folders/1aNXHyj-PU3hJcaofWqabRh3urmr4H_g-). We will also open source our description generation dataset and evaluation code.

---

### Decision · Program_Chairs · 2023-01-20

**Decision:**

Reject

**Justification For Why Not Higher Score:**

The paper has been discussed thoroughly during the response period, and most reviewers think the paper is not strong enough for the ICLR standards yet.

**Justification For Why Not Lower Score:**

N/A

**Metareview: Summary, Strengths And Weaknesses:**

This paper investigates the problem of using LLMs for understanding HTMLs. They consider three tasks: Semantic Classification of HTML elements, Description Generation for HTML inputs, and Autonomous Web Navigation of HTML pages and show encouraging results.

The paper has been discussed thoroughly during the discussion phase, and if my understanding is correct, several reviewers increased their evaluations after the authors’ response. However, the final consensus is that the paper is still below the bar for the following reasons (the scores are 5/5/5/6):
* Technical novelty and contributions are limited for conferences like ICLR.
* It lacks an in-depth comparison with HTML-specific pre-trained models, and it is not very clear how much structural information is needed for the tasks used in this paper
* It seems that the paper has room for improvement in terms of clarity and reproducibility.

Based on the above points, I can’t recommend acceptance in the current form.